# Collagen induces activation of DDR1 through lateral dimer association and phosphorylation between dimers

Victoria Juskaite, David S Corcoran, Birgit Leitinger*

National Heart and Lung Institute, Imperial College London, London, United Kingdom

**Abstract** The collagen-binding receptor tyrosine kinase DDR1 (discoidin domain receptor 1) is a drug target for a wide range of human diseases, but the molecular mechanism of DDR1 activation is poorly defined. Here we co-expressed different types of signalling-incompetent DDR1 mutants ('receiver') with functional DDR1 ('donor') and demonstrate phosphorylation of receiver DDR1 by donor DDR1 in response to collagen. Making use of enforced covalent DDR1 dimerisation, which does not affect receptor function, we show that receiver dimers are phosphorylated in trans by the donor; this process requires the kinase activity of the donor but not that of the receiver. The receiver ectodomain is not required, but phosphorylation in trans is abolished by mutation of the transmembrane domain. Finally, we show that mutant DDR1 that cannot bind collagen is recruited into DDR1 signalling clusters. Our results support an activation mechanism whereby collagen induces lateral association of DDR1 dimers and phosphorylation between dimers.

## Introduction

The discoidin domain receptor (DDR) subfamily of receptor tyrosine kinases (RTKs) comprises two members, DDR1 and DDR2. The DDRs regulate cell adhesion, cell migration and differentiation in a number of mammalian tissues (*Leitinger, 2014*). Both DDRs play key roles in embryo development: DDR1, for instance, is essential for mammary gland development (*Vogel et al., 2001*), while DDR2 mediates bone growth (*Ali et al., 2010*; *Bargal et al., 2009*; *Labrador et al., 2001*). The DDRs also play key roles in disease progression in a wide range of disorders including organ fibrosis, inflammation, osteoarthritis, atherosclerosis and many different types of cancer (*Borza and Pozzi, 2014*; *Leitinger, 2014*). Both DDRs are well-recognised drug targets but how ligand binding translates to DDR kinase activation has been poorly defined.

Uniquely among RTKs, the DDRs bind to key structural proteins found in all types of extracellular matrices, namely different types of collagen. Fibrillar collagens are ligands for both DDR1 and DDR2, while non-fibrillar collagens have different DDR preferences, with collagen IV exclusively binding to DDR1 and collagen X preferring DDR2 over DDR1 (*Leitinger, 2003*; *Leitinger and Kwan, 2006*; *Shrivastava et al., 1997*; *Vogel et al., 1997*). The interactions of the DDRs with fibrillar collagens are well understood: receptor binding sites have been mapped to specific amino acid motifs with the use of collagen-mimetic triple-helical peptides (*Konitsiotis et al., 2008*; *Xu et al., 2011*), and structural studies have revealed the details of the interactions (*Carafoli et al., 2009*; *Ichikawa et al., 2007*). In contrast, the nature of DDR binding sites on non-fibrillar collagens is currently not known.

Like all RTKs, the DDRs are composed of a ligand-binding extracellular region, a transmembrane domain and a cytoplasmic region that contains the catalytic kinase domain. DDR1 and DDR2 share a high degree of homology, in particular in their globular domains. The extracellular DDR region

*For correspondence:
b.leitinger@imperial.ac.uk

**Competing interests:** The authors declare that no competing interests exist.

**eLife digest** The membrane surrounding each living cell contains a variety of proteins that carry out different roles. For example, proteins called receptor tyrosine kinases help a cell to receive signals from its external environment. Receptor tyrosine kinases span the membrane so that one part of the protein known as the ectodomain sticks out from the surface of the cell, while another part (called the kinase domain) sits inside the cell.

When a signalling molecule binds to the ectodomain, the kinase domain becomes active and starts to add chemical groups called phosphates to other proteins. This process, known as phosphorylation, changes the protein's activity, which in turn influences the cell's behaviour. In most cases, the signalling molecule causes two receptor tyrosine kinase proteins to bind to each other and form a "dimer" in which the kinase domains are able to phosphorylate, and thus activate, each other.

Female mammals need a receptor tyrosine kinase called DDR1 to develop mammary glands (the glands that produce milk). DDR1 is activated when a signalling molecule called collagen binds to its ectodomain. Unlike many other receptor tyrosine kinases, DDR1 exists as a dimer even before it binds to collagen, so it is not clear how collagen activates DDR1. One possibility is that collagen causes several DDR1 dimers to form clusters on the membrane so that kinases on neighbouring dimers can phosphorylate each other. Juskaite et al. explored this idea by pairing up normal DDR1 proteins with mutant versions that are unable to bind to collagen.

The experiments show that when collagen binds to the normal DDR1 molecules, DDR1 dimers do indeed form clusters. This enables the normal protein molecules in neighbouring dimers to phosphorylate each other as well as the mutant proteins. In this way, the clustered DDR1 dimers can become active even if the clusters contain one or more mutant versions that are unable to detect collagen. Further experiments show that specific contacts need to form between neighbouring dimers for this phosphorylation to occur.

Abnormal DDR1 activity is associated with several diseases including cancer, inflammation and fibrosis. The findings of Juskaite et al. suggest that developing new drugs that can prevent DDR1 from forming clusters may help to treat people with these conditions. Further work is also needed to analyse the size and structure of DDR1 clusters and investigate if other proteins also associate with the clusters.

contains two globular domains: an N-terminal, ligand-binding discoidin domain that is tightly linked to a discoidin-like domain (*Leitinger, 2014*). These globular domains are followed by a highly flexible extracellular juxtamembrane region rich in glycine and proline residues (*Xu et al., 2014*). The intracellular DDR regions contain unusually large juxtamembrane regions (up to 171 amino acids in DDR1, depending on the isoform, and 142 amino acids in DDR2) that are followed by a catalytic kinase domain lacking a C-terminal tail (*Leitinger, 2014*). The collagen-binding site of DDRs is entirely contained in the discoidin domain (*Ichikawa et al., 2007*; *Leitinger, 2003*), which forms an 8-stranded $\beta$-barrel structure (*Carafoli et al., 2009*, *2012*; *Ichikawa et al., 2007*). A conserved trench at the top of the discoidin domain accommodates the collagen triple helix. The structures of the collagen-bound DDR2 discoidin domain (*Carafoli et al., 2009*) and that of the DDR1 discoidin domain in the absence of ligand (*Carafoli et al., 2012*) are very similar and do not provide any information on how ligand binding results in intracellular kinase activation. This is in sharp contrast to the situation of collagen-binding integrins where large conformational changes in the ligand-binding domain are transmitted to the rest of the receptor to drive transmembrane signalling (*Luo et al., 2007*).

A crystal structure of the inactive DDR1 kinase revealed a typical bilobal architecture, with kinase autoinhibition through interactions of the activation loop with the active site (*Canning et al., 2014*). The DDR cytoplasmic regions contain several sites of potential tyrosine phosphorylation, both in the juxtamembrane regions and in the kinase domains (*Leitinger, 2014*; *Lemeer et al., 2012*). While ligand-induced activation loop phosphorylation of the kinase domain has been demonstrated experimentally (*Iwai et al., 2013*; *Lemeer et al., 2012*; *Xu et al., 2014*), only a few juxtamembrane

tyrosines have been shown to be phosphorylated in response to collagen binding to cells (*Iwai et al., 2013*; *Lemeer et al., 2012*). Moreover, little is known about signalling pathways that are initiated by the phosphorylation of particular DDR tyrosine residues (*Leitinger, 2014*).

The DDRs differ from typical RTKs by several unusual features, the mechanistic bases of which are poorly understood. One of their puzzling characteristics is the slow kinetics of collagen-induced autophosphorylation, which in some cell types can take hours to reach a maximum and then remains detectable for more than a day (*Shrivastava et al., 1997*; *Vogel et al., 1997*). Another distinguishing feature is that the DDRs form stable, non-covalent dimers in the absence of ligand (*Mihai et al., 2009*; *Noordeen et al., 2006*; *Xu et al., 2014*). A key dimerisation interface is located in the DDR transmembrane domain via a leucine-based sequence motif; however, multiple receptor-receptor contacts in the extracellular and cytosolic regions also seem to contribute to DDR dimerisation (*Noordeen et al., 2006*). The isolated DDR1 and DDR2 transmembrane helices interact very strongly, as detected in a bacterial TOXCAT reporter assay (*Noordeen et al., 2006*). In fact, a systematic study that compared the self-interaction potential of all RTK transmembrane domains found that the DDR1 and DDR2 transmembrane domains gave the strongest signal of all RTKs in this assay (*Finger et al., 2009*). In previous work, we performed cysteine-scanning mutagenesis of the 10 extracellular juxtamembrane residues closest to the DDR1 transmembrane domain and found that most of the mutant constructs were crosslinked with nearly 100% efficiency, which led us to conclude that DDR1 dimerisation is constitutive, rather than in a dynamic equilibrium with monomers (*Xu et al., 2014*). We further concluded that dimerisation occurs during biosynthesis (*Noordeen et al., 2006*; *Xu et al., 2014*). While ligand-independent non-covalent dimerisation has been observed for other RTKs, including the EGF receptor family, these RTKs exist in a dynamic equilibrium between monomers and inactive dimers (discussed in [*Maruyama, 2014*]).

Because the DDRs are constitutively dimerised in the absence of ligand, the paradigm of ligand-induced RTK dimerisation (*Lemmon and Schlessinger, 2010*) cannot apply to them. The insulin receptor is a covalently-linked dimer that is activated by ligand-induced conformational changes within the dimer (*Kavran et al., 2014*; *Ward et al., 2013*). However, structural studies found no evidence for ligand-induced conformational changes in the DDR ectodomains (see above; [*Carafoli et al., 2009*, *2012*; *Ichikawa et al., 2007*]), and biochemical experiments led to the conclusion that transmembrane signalling of DDR1 occurs without coupling through the flexible juxtamembrane region (*Xu et al., 2014*). Together, these studies ruled out a mechanism of signalling within DDR1 dimers and led to the hypothesis that DDR1 activation may be associated with ligand-induced clustering (lateral association) of receptors in the cell membrane (*Xu et al., 2014*).

Here we show evidence for a mechanism that is compatible with ligand-induced clustering of DDR1 dimers. Co-expression studies using a range of signalling-incompetent DDR1 mutants with different forms of signalling-competent DDR1 and DDR2 showed that collagen-induced DDR1 activation involves phosphorylation in trans between neighbouring DDR1 dimers. Phosphorylation between DDR1 dimers occurs both on the juxtamembrane region and the kinase activation loop and can be elicited by different types of ligand. Moreover, phosphorylation in trans requires specific contacts between the transmembrane domains, but not between the ectodomains. These findings define a unique mechanism of transmembrane signalling by DDR1.

## Results

### Experimental approach

In this study we explored the mechanism of ligand-induced phosphorylation of DDR1 by co-expression of signalling-defective DDR1 mutants with signalling-competent DDR1. Receptor phosphorylation was visualised by Western blotting with a phospho-specific antibody (Ab) against a key tyrosine phosphorylation site in the intracellular juxtamembrane region, tyrosine-513 (*Xu et al., 2014*). DDR1 has two main isoforms, DDR1a and DDR1b, with DDR1b containing an additional 37 amino acids in its cytosolic juxtamembrane region (*Leitinger, 2014*). Tyrosine-513 within this alternatively spliced insert is one of the few experimentally verified ligand-induced tyrosine phosphorylation sites in the juxtamembrane region (*Lemeer et al., 2012*).

The signalling-defective DDR1 mutants had mutations in the ectodomains but intact cytosolic juxtamembrane regions and kinase domains. As detailed below, co-expression with signalling-

competent DDR1 resulted in ligand-induced phosphorylation of the signalling-defective mutants, which on their own could not be phosphorylated in response to collagen. Hence, in our co-expression experiments, we use the terms 'donor kinase' for the signalling-competent DDR1 and 'receiver kinase' for the signalling-defective mutants, respectively. In order to distinguish the cytoplasmic domains of receiver and donor DDR1 we replaced tyrosine-513 in the donor receptor with a phenylalanine, thereby creating DDR1b-Y513F, which cannot be phosphorylated at this residue (*Figure 1A*). This allowed us to monitor exclusively the phosphorylation of receiver DDR1 by Western blotting with the Ab against phosphorylated tyrosine-513.

## Ligand-induced phosphorylation between DDR1 dimers

We first confirmed expression and collagen-induced activation of DDR1b-Y513F in transiently transfected HEK293 cells. Collagen-induced autophosphorylation of the activation loop (anti-pY792) was detected at similar levels as in wild-type DDR1b, whereas, as expected, no signal was detectable for DDR1b-Y513F with the Ab against phosphorylated tyrosine-513 (anti-pY513; *Figure 1B*). Mutation of a key ligand-binding residue in DDR2, tryptophan-52 (conserved in DDR1 where it corresponds to tryptophan-53), severely reduces collagen binding affinity (*Carafoli et al., 2009*; *Ichikawa et al., 2007*) and renders the full-length receptor unable to become activated by collagen (*Carafoli et al., 2009*). We created DDR1b-W53A as a ligand binding-defective DDR1 construct, which, on its own, does not respond to collagen with increased autophosphorylation. Co-expression with DDR1b-Y513F, however, led to collagen I-induced phosphorylation of tyrosine-513 in DDR1-W53A (*Figure 1C*). In a previous study we identified a conserved patch on the surface of DDR1 that is required for transmembrane signalling seemingly without participating in ligand binding (*Carafoli et al., 2012*). Co-expression of the signal-patch mutants, DDR1b-R32E or DDR1b-L152E, with DDR1b-Y513F restored collagen-induced phosphorylation of tyrosine-513 in the signal-patch mutants (*Figure 1C*). Rescue of DDR1b-W53A phosphorylation through co-expression with DDR1b-Y513F was not restricted to activation by fibrillar collagen I, but could also be induced by binding to the network-forming collagen IV (*Figure 1D*).

Phosphorylation of signalling-defective receiver DDR1 mutants in the co-expression system could arise through transphosphorylation within heterodimers of receiver and donor receptors and/or phosphorylation between dimers in trans. To investigate whether the receiver mutants were phosphorylated as a dimer, we created covalently-linked homodimers of the DDR1 receiver mutants. Cysteine replacement of threonine-416, a residue in the extracellular DDR1 juxtamembrane region close to the transmembrane domain, results in covalently cross-linked DDR1 dimers via a disulphide bond at residue 416, without affecting collagen-induced receptor activation (*Xu et al., 2014*). Non-reducing SDS-PAGE followed by Western blotting of cell lysates from cells expressing DDR1b-W53A/T416C, DDR1b-R32E/T416C or DDR1b-L152E/T416C (referred to as DDR1b-W53A-Cys, DDR1b-R32E-Cys and DDR1b-L152E-Cys from here on), showed that the mature forms of the double mutants were efficiently dimerised, as expected (*Figure 2A and B*, anti-DDR1 blot, lanes 7 and 8). Co-expression of any of the dimerised signalling-defective receiver mutants with DDR1b-Y513F resulted in collagen-induced phosphorylation at the position of the disulphide-linked dimers (*Figure 2*, see also *Figure 2—figure supplement 1*), demonstrating conclusively that collagen induces phosphorylation in trans between DDR1 dimers.

## Signalling between dimers is independent of DDR1 ectodomain contacts but requires specific transmembrane domain contacts

We next asked whether extracellular contacts were required for signalling between DDR1 dimers. Deletion of the entire DDR1 ectodomain (except for the seven amino acids closest to the transmembrane domain) did not impair the ability of DDR1b-Y513F to promote collagen-induced phosphorylation of the ectodomain deletion mutant (*Figure 3A*, left hand side). Cysteine replacement of T416 (full-length DDR1b numbering) in the deletion mutant resulted in a construct that was predominantly dimeric, but in contrast to the full-length dimerised constructs, a proportion of monomeric protein was also detectable (*Figure 3A*, bottom blot, lanes 7 and 8). Interestingly, co-expression with DDR1b-Y513F led to collagen-induced phosphorylation only of the truncated dimer, with very little phosphorylated monomer detectable (*Figure 3A*, top right blot).

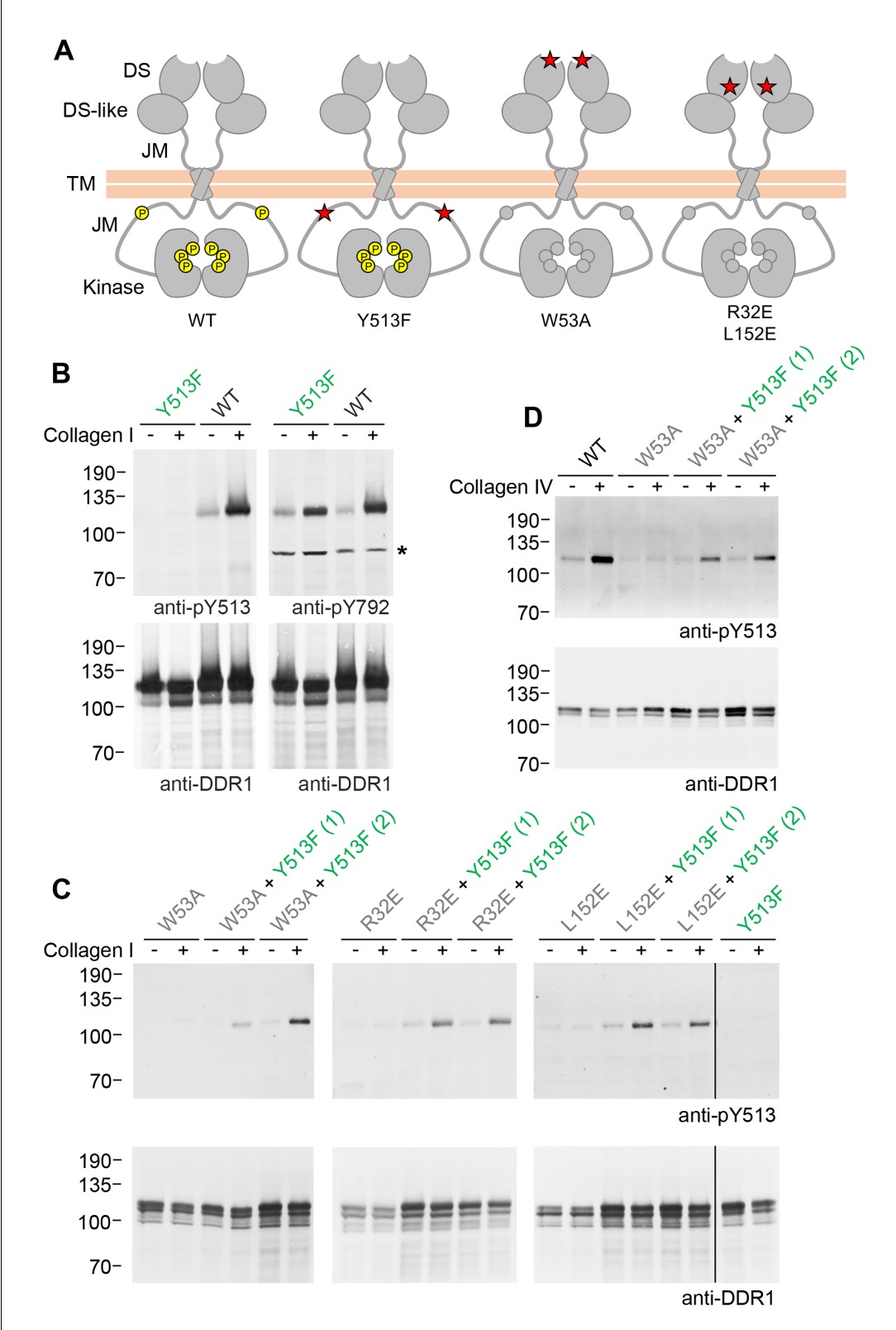

**Figure 1.** Co-expression of DDR1 donor kinase with signalling-incompetent receiver DDR1 mutants leads to collagen-induced phosphorylation of receiver DDR1. (A) Schematic diagrams of wild-type and mutant DDR1b. The extracellular region consists of the ligand-binding discoidin (DS) domain, the discoidin-like (DS-like) domain and a flexible juxtamembrane (JM) region. The transmembrane (TM) domain contains a dimerising motif. The cytoplasmic region contains a large unstructured JM region, followed by the catalytic kinase domain. Collagen binding induces phosphorylation of

*Figure 1 continued on next page*

*Figure 1 continued*

DDR1b on cytoplasmic tyrosine residues; Y513 in the JM region and the three activation loop tyrosines are shown in phosphorylated form for WT DDR1b as yellow circles. DDR1b-Y513F is phosphorylated on the activation loop but cannot be phosphorylated on Y513. Ligand-binding defective DDR1b-W53A and signalling-defective DDR1b-R32E or DDR1b-L152E are DDR1 mutants that are not phosphorylated upon collagen incubation. (**B–D**) Wild-type or mutant DDR1b constructs were transiently expressed in HEK293 cells, either alone, or co-expressed as indicated. Cells were stimulated with collagen I or collagen IV for 90 min at 37°C. Aliquots of cell lysates were analysed by reducing SDS-PAGE and Western blotting. The blots were probed with phospho-specific Abs, as indicated, and re-probed with anti-DDR1. *, non-specific band. (**C, D**) Co-expression was performed with different amounts of DDR1b-Y513F expression vector, with the higher amount denoted by (2). The positions of molecular mass markers are indicated on the left (in KDa).

Because our experimental approach relies on co-expression of mutant DDR1 constructs, our experiments can only be done under transfection conditions. A potential concern is that over-expression may lead to artificial DDR1 aggregates, which are responsible for phosphorylation between dimers. Flow cytometry of DDR1 showed that under standard transfection conditions used for co-expression experiments, mean DDR1 fluorescence levels were similar to those on the breast cancer cell line T47D but approximately three times higher than those on primary keratinocytes (*Figure 3—figure supplement 1*; HEK 5 µg DNA vs T47D). To exclude that phosphorylation between dimers is dependent on high DDR1 expression levels, we performed co-expression experiments with lower DDR1 expression levels and used the most sensitive chemiluminescence substrate for detection of tyrosine-513 phosphorylation. Ligand-induced dimer phosphorylation was detected for both DDR1-R32-Cys and DDR1-ΔECD-Cys at a range of expression levels including those that would lead to surface levels comparable to those on keratinocytes (*Figure 3—figure supplement 2*).

In a previous study we identified a leucine-based sequence motif in the DDR1 transmembrane domain that is required for strong transmembrane helix association, as well as for collagen-induced receptor activation (*Noordeen et al., 2006*). In order to analyse whether transmembrane helix interactions play a role in signalling between DDR1 dimers, we introduced the inactivating transmembrane mutation (L430G/L431P, termed TM1) into the ectodomain deletion constructs. Surface immunostaining and flow cytometry confirmed that the TM1 mutants were present on the cell surface at comparable levels with the respective constructs with wild-type transmembrane regions (*Figure 4—figure supplement 1*), demonstrating that the transmembrane mutations did not affect protein trafficking. However, these constructs could no longer be phosphorylated through co-expression with DDR1b-Y513F, despite the presence of cysteine-linked dimers in the ΔECD-Cys-TM1 construct (*Figure 4*). These results indicate that specific interactions through the DDR1 transmembrane domain are required for phosphorylation between dimers.

## Signalling between dimers is not dependent on intracellular symmetry

We next explored whether signalling between dimers required the co-expressed DDR1 constructs to have identical juxtamembrane or kinase domains. Co-expression of receiver DDR1b constructs with DDR1a (which lacks the 37 amino acid segment that contains tyrosine-513) resulted in efficient collagen-induced tyrosine-513 phosphorylation of the signalling-incompetent receiver dimers (*Figure 5*). A more drastic test was to replace the donor kinase with the homologous DDR2. The kinase domains of the DDRs are highly conserved (~67% identity), but the intracellular juxtamembrane regions are much less conserved (~30% identity between DDR1b and DDR2). Co-expression with DDR2 as the donor kinase similarly led to phosphorylation of the DDR1b receiver mutants (*Figure 5*, see also *Figure 5—figure supplement 1*). Therefore, phosphorylation between DDR dimers in trans does not have a strict requirement for a particular juxtamembrane sequence.

## Signalling between dimers does not require kinase activity of the receiver kinase

So far we monitored activation of DDR1 mutant receiver dimers by tyrosine-513 phosphorylation. Blotting with phospho-specific Abs against the activation loop tyrosines, tyrosine-792 or tyrosine-796, showed that DDR1 receiver dimers were also phosphorylated on the activation loop (*Figure 6A,B*, see also *Figure 6—figure supplement 1*), indicating that co-expression with functional DDR1 donor kinase results in enhanced kinase activity of the receiver kinase. We therefore

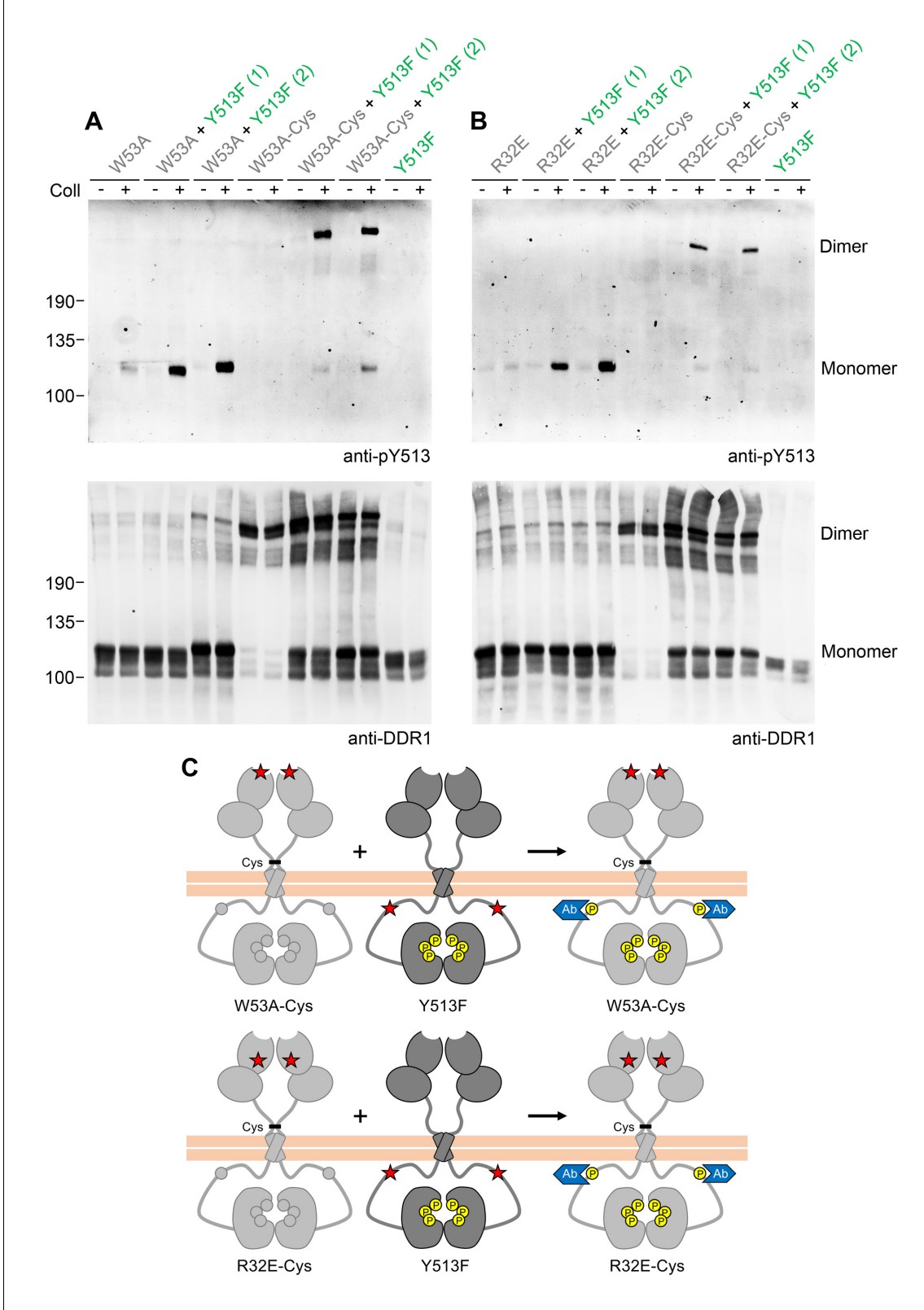

**Figure 2.** Co-expression with DDR1 donor kinase results in ligand-induced phosphorylation of receiver DDR1 dimers. (**A**, **B**) Ligand-binding defective (W53A or W53A-Cys) or signalling-defective (R32E or R32E-Cys) DDR1b constructs were transiently expressed in HEK293 cells, either alone or co-expressed with DDR1b-Y513F, as indicated. Cells were stimulated with collagen I (Coll) for 90 min at 37°C or left untreated. Cells were lysed in the presence of NEM, and aliquots of cell lysates were analysed by non-reducing SDS-PAGE followed by Western blotting. The blots were probed with

*Figure 2 continued on next page*

*Figure 2 continued*
phospho-specific anti-pY513 (upper blots), and re-probed with anti-DDR1 (lower blots). Co-expression was performed with different amounts of DDR1b-Y513F expression vector, with the higher amount denoted by (2). The positions of molecular mass markers are indicated on the left (in KDa). The positions of cysteine-linked dimeric DDR1 and of mature monomeric DDR1 are indicated on the right. (C) Schematic diagram showing that co-expression of dimeric receiver DDR1b (W53A-Cys or R32E-Cys) with donor DDR1b-Y513F leads to collagen-induced Y513 phosphorylation of receiver dimers. The blue shape denotes anti-pY513 used for Western blotting.
The following figure supplement is available for figure 2:

**Figure supplement 1.** Co-expression with DDR1 donor kinase results in ligand-induced phosphorylation of DDR1-L152E dimers.

analysed whether autocatalytic kinase activity of the receiver kinase was required for tyrosine-513 or activation loop phosphorylation when co-expressed with donor DDR1. Mutation of lysine-655 (DDR1b numbering) in the catalytic domain of DDR1 abolishes collagen-induced DDR1 phosphorylation (*Vogel et al., 2000*). We co-expressed DDR1b-K655A (hereafter referred to as DDR1b-KD), with or without the T416C mutation, with functional donor DDR1 and observed efficient ligand-induced tyrosine-513 phosphorylation of both kinase-dead receiver constructs (*Figure 6C*). Furthermore, collagen-induced activation loop phosphorylation of dimeric DDR1-Cys-KD was also observed when co-expressed with functional donor DDR1 (*Figure 6—figure supplement 2*), demonstrating receiver activation loop phosphorylation in the absence of receiver kinase activity. These data show that phosphorylation between dimers can occur independently of catalytic activity of the receiver kinase and suggest that the receiver kinase dimer functions as a substrate of the functional donor receptor (DDR1a, DDR1-Y513F or DDR2), which acts as an enzyme kinase in our co-expression experiments.

## Signalling between dimers requires kinase activity of the donor kinase

The requirement for kinase activity of the donor kinase was addressed by co-expression of the ecto-domain deletion dimer, DDR1b-ΔECD-Cys, as the receiver kinase, with the kinase-dead form of DDR1a, DDR1a-KD, as the donor kinase (*Figure 7*). Neither DDR1a-KD nor a truncated version of DDR1b, DDR1-MDN, which retains part of the DDR1b juxtamembrane region but lacks the kinase domain (*Noordeen et al., 2006*), were able to act as donor kinases and induce phosphorylation of the receiver construct, demonstrating that catalytic activity of the donor kinase is necessary for phosphorylation between dimers.

## Ligand-independent activation by a multimeric antibody leads to signalling between dimers

In addition to stimulation by authentic collagen ligands, DDR1 phosphorylation can also be induced by an activating Ab, mAb-513 (*Figure 8A*). This monoclonal Ab is a multimeric IgM Ab. Since mAb-513 binds to an extracellular epitope, the ectodomain deletion mutant, DDR1b-ΔECD-Cys, cannot bind this Ab and hence was not phosphorylated by incubation of cells with mAb-513 (*Figure 8B*). Co-expression with DDR1b-Y513F, however, led to mAb-513-induced phosphorylation of DDR1b-ΔECD-Cys dimers. Similar to the situation with collagen-induced phosphorylation of the ectodomain deletion construct by DDR1-Y513F, mAb-513-induced phosphorylation was dependent on the trans-membrane domain of the receiver kinase (*Figure 8B,C*). These data show that phosphorylation between dimers is independent of the specific activation mode of the donor kinase.

## Signalling between dimers is not dependent on collagen multivalency

Fibrillar collagens are triple-helical proteins that assemble into fibrillar structures in tissues. In our co-expression experiments we incubated cells with soluble collagen I, but we cannot rule out that small fibrils form during the 90 min incubation times. These fibrils could provide multiple binding sites for DDR1, thereby clustering the receptor. We previously showed that DDR1 activation could be induced by collagen-mimetic synthetic triple-helical peptides encompassing a DDR1 binding motif (*An et al., 2016*; *Xu et al., 2011*). Because these peptides are not able to form higher-order aggregates, we concluded that DDR phosphorylation can be induced by single collagen triple helices. Co-expression of DDR1b-ΔECD-Cys with DDR1b-Y513F showed that a collagen-mimetic DDR1 peptide

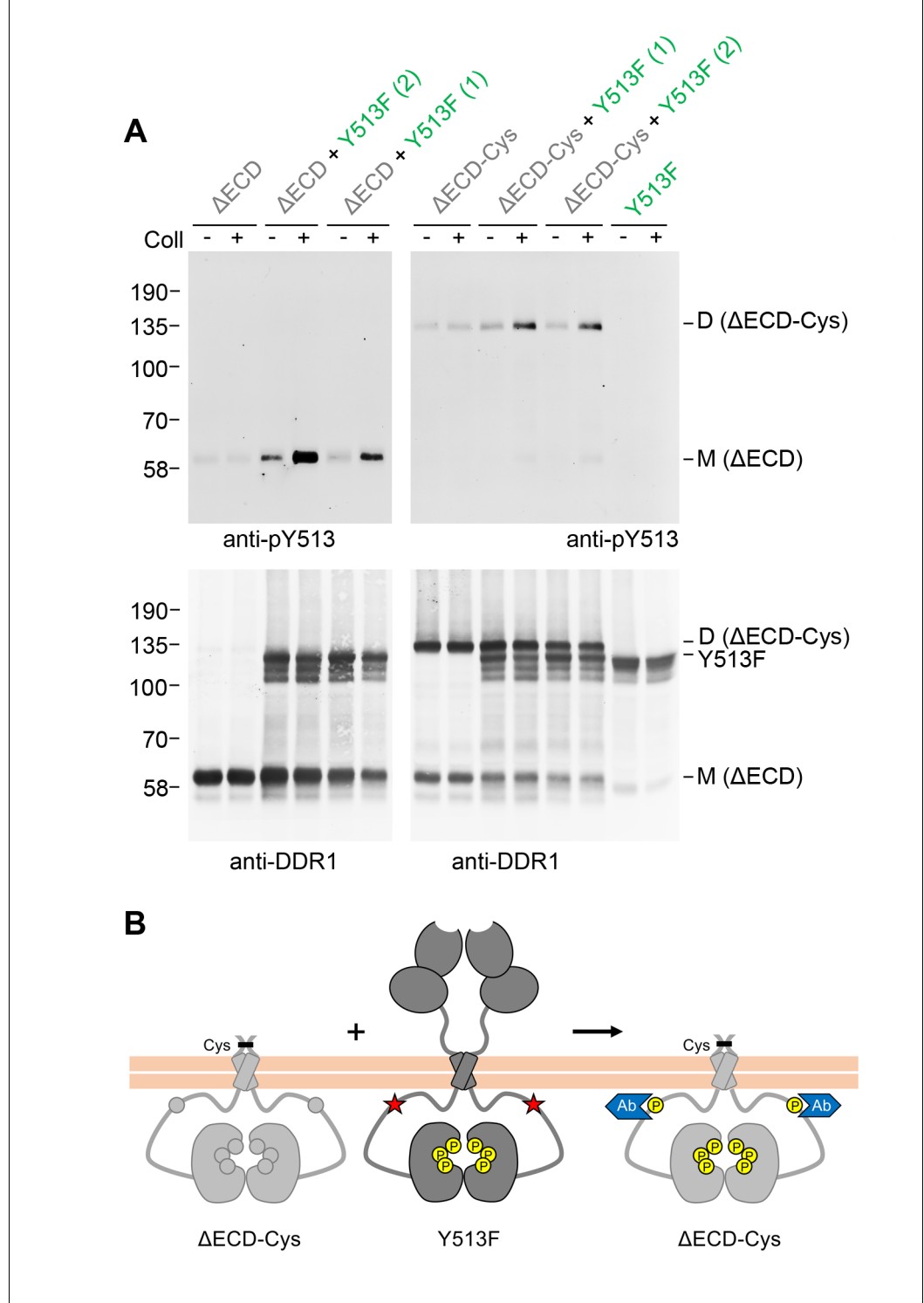

**Figure 3.** Signalling between DDR1 dimers is independent of ectodomain contacts. (**A**) Ectodomain-deletion constructs (DDR1b-ΔECD or DDR1b-ΔECD-Cys) were transiently expressed in HEK293 cells, either alone or co-expressed with DDR1b-Y513F, as indicated. The cells were stimulated with collagen I (Coll) for 90 min at 37°C or left untreated. Cells were lysed in the presence of NEM, and aliquots of cell lysates were analysed by non-reducing SDS-PAGE followed by Western blotting. The blots were probed with phospho-specific anti-pY513 (upper blots), and re-probed with anti-DDR1 (lower blots). Co-expression was performed with different amounts of DDR1b-Y513F expression vector, with the higher amount denoted by (2). The positions of molecular mass markers are indicated on the left (in KDa). The positions of cysteine-linked dimeric DDR1b-ΔECD-Cys, DDR1b-Y513F and

*Figure 3 continued on next page*

*Figure 3 continued*

mature monomeric DDR1b-*Δ*ECD are indicated on the right. (**B**) Schematic diagram showing that co-expression of ectodomain-deleted dimeric DDR1b with donor DDR1b-Y513F leads to collagen-induced Y513 phosphorylation of ectodomain-deletion dimers. The blue shape denotes anti-pY513 used for Western blotting.

The following figure supplements are available for figure 3:

**Figure supplement 1.** Cell surface and total protein DDR1 expression in DDR1-overexpressing HEK293 cells, primary keratinocytes and cell lines.

**Figure supplement 2.** Co-expression with DDR1 donor kinase results in ligand-induced phosphorylation of receiver DDR1 dimers even at low expression levels.

ligand could also induce phosphorylation between dimers (*Figure 9*), demonstrating that ligand multivalency is not required for signalling between dimers.

## Collagen binding results in redistribution of DDR1 on the cell surface

Phosphorylation between DDR1 dimers requires close apposition of DDR1 molecules, which is likely a result of ligand-induced receptor clustering. We used immunofluorescence to study collagen-induced DDR1 redistribution on the cell surface. *Figure 10* demonstrates that DDR1b on the surface of unstimulated Cos-7 cells is present in a punctate distribution. Collagen binding led to redistribution of cell-surface DDR1 into a more compact structure, a phenomenon that is consistent with ligand-induced clustering. Ligand-induced clustering was prevented when cells were incubated with collagen in the presence of a function-blocking anti-DDR1 mAb. This mAb binds to an extracellular epitope and allosterically blocks collagen-induced DDR1 phosphorylation (*Carafoli et al., 2012*). Hence, collagen-induced clustering is a key step in DDR1 activation.

## Signalling between dimers leads to local receptor phosphorylation only

Eph receptors are RTKs that require clustering by ligand for receptor activation (*Nikolov et al., 2014*). It was previously reported that Eph signalling clusters extend beyond the direct receptor-ligand contact area, and phosphorylated receptors were detectable over an extensive area outside the ligand-contact zone (*Wimmer-Kleikamp et al., 2004*). To study the localisation of DDR1 phosphorylation with respect to ligand contact surfaces, we used collagen I-coated beads. DDR1b was expressed in Cos-7 cells, and phosphorylation was visualised with immunofluorescence. The beads efficiently recruited DDR1, which was phosphorylated (*Figure 11A*). However, in contrast to the situation with Eph receptors, phosphorylated DDR1 was strictly limited to the bead contact areas, even after prolonged (4 hr) incubation times.

## The ectodomain-deletion construct is efficiently recruited to the ligand contact zone when co-expressed with functional DDR1

DDR1 lacking its ectodomain cannot bind ligand. Expression of DDR1b-*Δ*ECD in Cos-7 cells did not result in DDR1b-*Δ*ECD recruitment to collagen-coated beads, as expected (*Figure 11B*). However, co-expression of DDR1b-*Δ*ECD with DDR1-Y513F showed efficient recruitment of DDR1b-*Δ*ECD into the bead contact area, as well as Y513 phosphorylation that was restricted to the bead area (*Figure 11B*). These data strongly support the concept that ligand binding leads to lateral association of DDR1 dimers.

## Discussion

Here we demonstrate an activation mechanism for DDR1 that is distinct from that of all other RTK families. Previous work showed that DDR1 forms constitutive dimers (or higher oligomers) in the absence of ligand (*Mihai et al., 2009*; *Noordeen et al., 2006*; *Xu et al., 2014*) ruling out the widely accepted RTK activation model of ligand-induced dimerisation (*Lemmon and Schlessinger, 2010*). Structural studies did not detect conformational differences between ligand-bound and unliganded DDR discoidin domains (*Carafoli et al., 2009*, *2012*; *Ichikawa et al., 2007*). Moreover, the long and

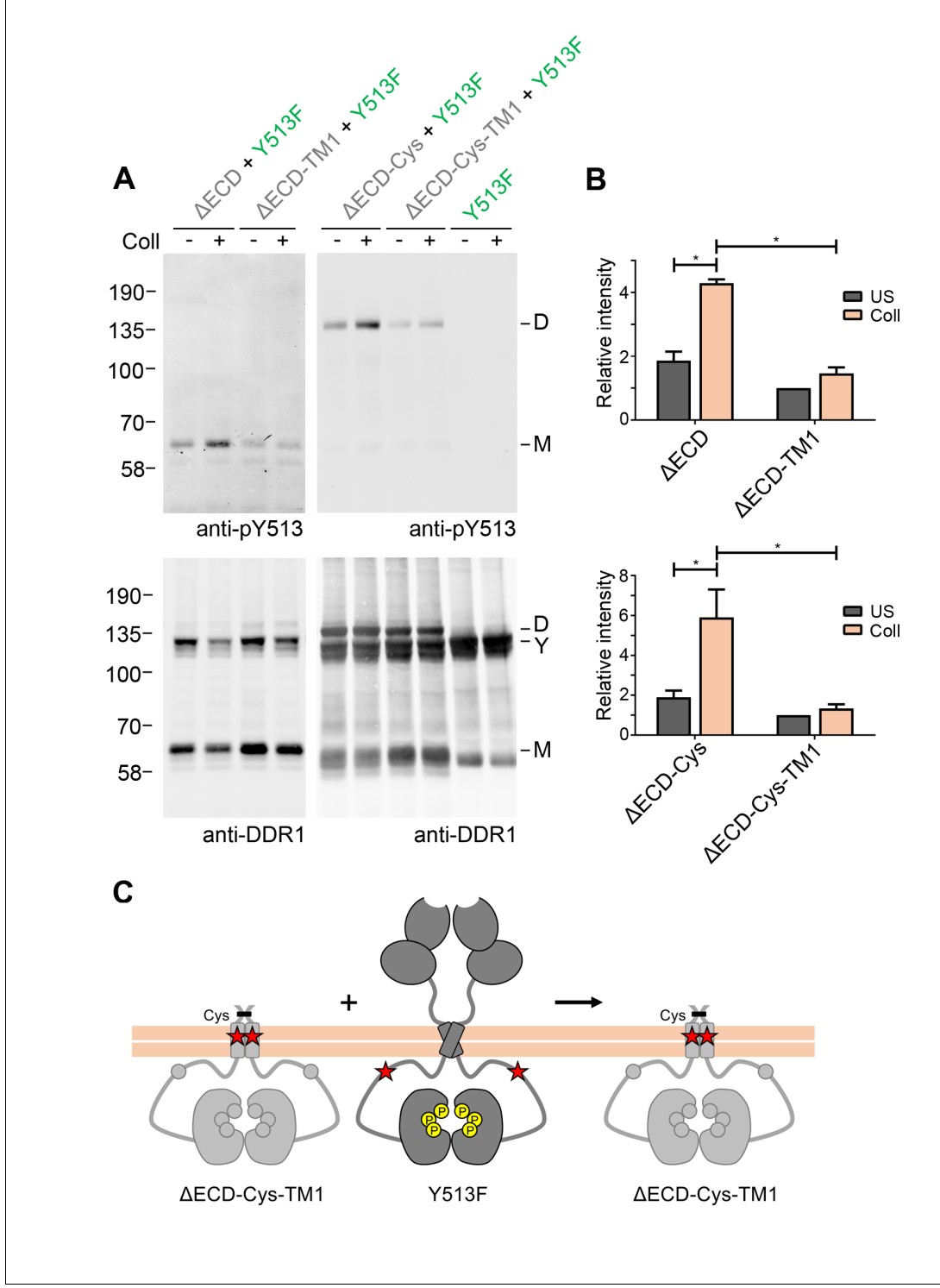

**Figure 4.** Signalling between DDR1 dimers is dependent on transmembrane domain contacts. (**A**) Ectodomain-deletion constructs (DDR1b-*ΔECD*, DDR1b-*ΔECD*-TM1, DDR1b-*ΔECD*-Cys or DDR1b-*ΔECD*-Cys-TM1) were transiently expressed in HEK293 cells, either alone or co-expressed with DDR1b-Y513F, as indicated. The cells were stimulated with collagen I (Coll) for 90 min at 37°C or left untreated. Cells were lysed in the presence of NEM, and aliquots of cell lysates were analysed by non-reducing SDS-PAGE followed by Western blotting. The blots were probed with phospho-specific anti-pY513 (upper blots), and re-probed with anti-DDR1 (lower blots). The positions of molecular mass markers are indicated on the left (in KDa). The positions of cysteine-linked dimeric DDR1b-*ΔECD*-Cys (D), of DDR1b-Y513F (Y) and of mature monomeric DDR1b-*ΔECD* (M) are indicated on the

*Figure 4 continued on next page*

*Figure 4 continued*

right. (**B**) Quantitation of receiver DDR1 pY513 signals (co-expressed with Y513F), normalised to respective DDR1 signals, expressed as relative band intensity with respect to the lowest signals on the blots (unstimulated DDR1b-ΔECD-TM1 or unstimulated DDR1b-ΔECD-Cys-TM1). US, unstimulated; Coll, stimulation with collagen I. *$p<0.05$; n = 4. (**C**) Schematic diagram showing that co-expression of ectodomain-deleted dimeric DDR1b with a transmembrane domain mutation and donor DDR1b-Y513F does not result in collagen-induced Y513 phosphorylation of the deletion construct.

The following figure supplement is available for figure 4:

**Figure supplement 1.** Cell surface expression of ectodomain-deletion constructs.

flexible extracellular DDR1 juxtamembrane region does not appear to couple conformational changes to the intracellular region (*Xu et al., 2014*). These observations led to the hypothesis that collagen binding may induce clustering of DDR1 in the cell membrane. In the present study we show that collagen binding results in DDR1 redistribution into a more compact structure on the cell surface. Furthermore, we demonstrate that ligand binding induces phosphorylation in trans between DDR1 dimers (as opposed to phosphorylation exclusively within a dimer), which can only occur if neighbouring dimers are brought into close proximity. These data, therefore, strongly support a clustering mechanism of DDR1 activation.

Our experiments with disulphide-linked constructs clearly demonstrate that inactive DDR1 mutants can be phosphorylated in a dimeric state. Phosphorylation of a key juxtamembrane tyrosine in the receiver construct (Y513) requires kinase activity of the donor but not of the receiver kinase. These data indicate a mechanism where the juxtamembrane regions of DDR1 dimers act as substrates for the kinase activity of neighbouring DDR1 molecules, rather than being phosphorylated exclusively through kinase activity within the same dimer. In addition to juxtamembrane phosphorylation, the activation loop of the receiver kinase was phosphorylated in our co-expression studies, again independently of receiver kinase activity, indicating that the activation loop can also act as a substrate for neighbouring DDR1 dimers. Activation loop phosphorylation is a major regulatory mechanism that releases cis-autoinhibition of kinases, including the majority of RTKs (*Lemmon and Schlessinger, 2010*). The DDR1 kinase domain is auto-inhibited through activation loop interactions with the active site (*Canning et al., 2014*). Our collective biochemical and microscopic data suggest that DDRs are activated by a mechanism in which collagen binding promotes lateral association of DDR1 dimers, which contributes to activation loop phosphorylation and perhaps also to the release of cis-autoinhibition of the kinase domain.

Co-clustering with, and phosphorylation by, catalytically active DDR1 was independent of ectodomain contacts but required specific transmembrane domain interactions. These data reinforce a key role for the DDR1 transmembrane domain in signalling. DDR1 transmembrane helices have the highest propensity for self-association amongst all RTKs (*Finger et al., 2009*), and DDR1 mutants with mutations that weaken transmembrane helix interactions are severely compromised in collagen-induced receptor activation (*Noordeen et al., 2006*). The present data show that intact transmembrane associations are not only essential for DDR1 as an enzyme kinase but also as a substrate kinase. It is possible that weakened transmembrane domain contacts affect the conformation of the cytoplasmic region in such a way that it no longer functions as a substrate for donor DDR1 kinase. Transmembrane domain contacts may also be required for DDR1 clustering, which could either involve direct receptor-receptor contacts or another membrane protein. In breast cancer cells, it was recently shown that the tetraspanin TM4SF1 promotes clustering of DDR1; however, in this scenario the DDR1 ectodomain was necessary for interaction with TM4SF1, while the transmembrane and cytoplasmic regions were dispensable (*Gao et al., 2016*).

DDR1 activation in our co-expression experiments was achieved with fibrillar and non-fibrillar collagens, a collagen-mimetic peptide, as well as by clustering with a multimeric anti-DDR1 mAb. The fact that a short collagen-mimetic peptide efficiently induced signalling between DDR1 dimers suggests that DDR1 clustering is receptor-mediated, rather than achieved through cross-linking via the ligand. While most RTKs are thought to signal as dimers, ligand-induced clustering is a key step in signalling of the Eph family of RTKs (*Himanen et al., 2010*; *Seiradake et al., 2010*). DDR1 activation

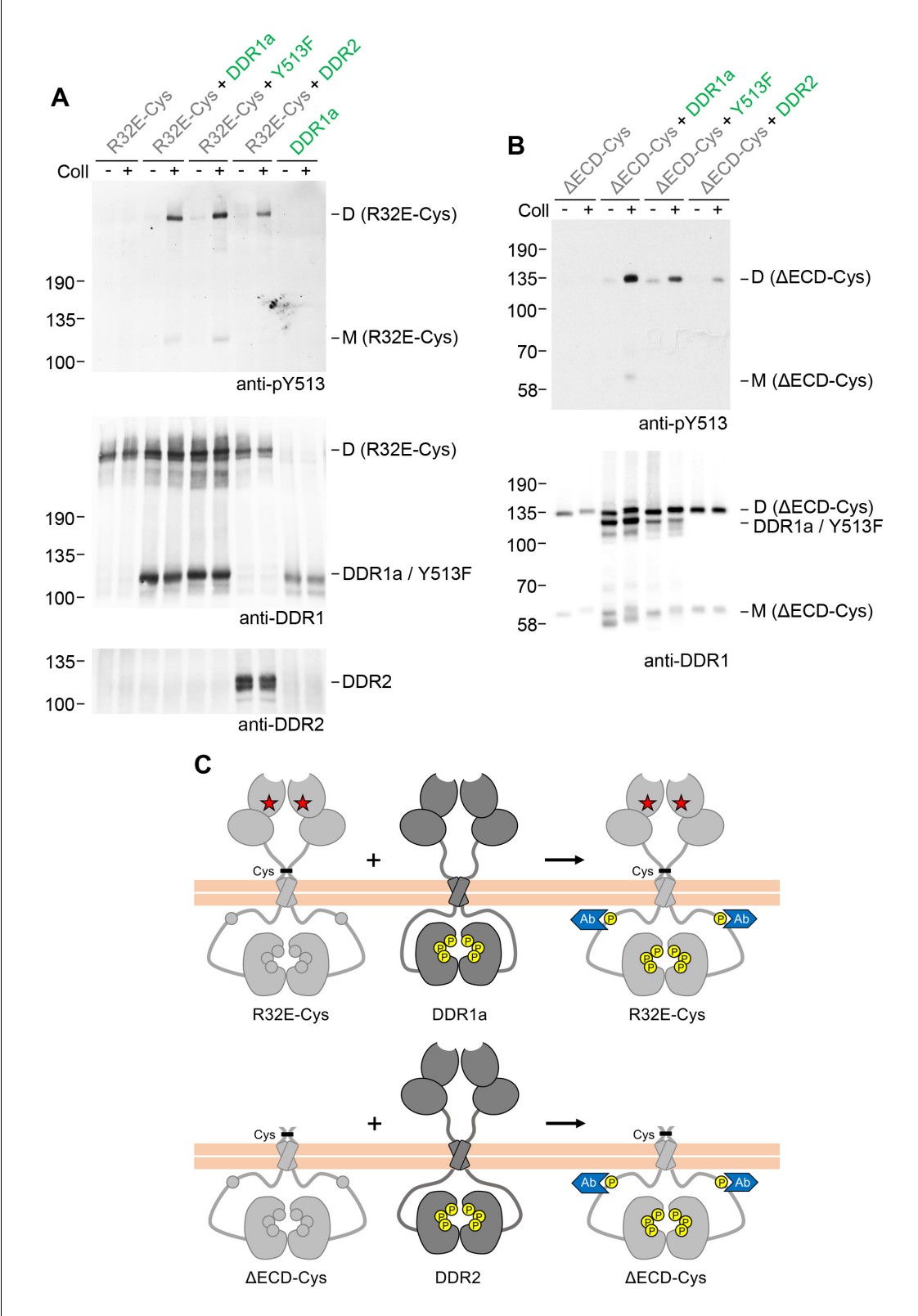

**Figure 5.** Signalling between dimers does not require identical cytoplasmic regions. (A, B) Signalling-defective (DDR1b-R32E-Cys) or ectodomain-deletion (DDR1b-ΔECD-Cys) constructs were transiently expressed in HEK293 cells, either alone or co-expressed with DDR1a, DDR1b-Y513F or DDR2, as indicated. The cells were stimulated with collagen I (Coll) for 90 min at 37°C or left untreated. Cells were lysed in the presence of NEM, and aliquots of cell lysates were analysed by non-reducing SDS-PAGE followed by Western blotting. The blots were probed with phospho-specific anti-pY513 (upper

*Figure 5 continued on next page*

*Figure 5 continued*

blots), and re-probed with anti-DDR1 (lower blots). The positions of molecular mass markers are indicated on the left (in KDa). The positions of cysteine-linked dimeric (D) or mature monomeric (M) DDR1 receiver constructs, as well as of DDR1a or DDR1b-Y513F donor DDR1 are indicated on the right. (**C**) Schematic diagram showing that co-expression of signalling-incompetent dimeric receiver DDR1 constructs with DDR1a (which lacks the region encompassing Y513) or DDR2 as the donor kinases results in collagen-induced Y513 phosphorylation of the receiver dimers. The blue shape denotes anti-pY513 used for Western blotting.

The following figure supplement is available for figure 5:

**Figure supplement 1.** Co-expression of signalling-incompetent receiver DDR1 mutants with DDR2 leads to collagen-induced phosphorylation of receiver mutants.

shares some, but not all of the characteristics of Eph receptor activation. Similar to what we show in this report for DDR1, ligand binding-defective Eph receptors co-cluster with functional Eph receptors (*Seiradake et al., 2013*; *Wimmer-Kleikamp et al., 2004*). Furthermore, kinase-dead EphB6 clusters with EphB1, resulting in EphB6 transphosphorylation (*Freywald et al., 2002*). However, unlike in DDR1 clusters, structurally defined ectodomain contacts are important for Eph receptor clusters (*Himanen et al., 2010*; *Seiradake et al., 2010*, *2013*; *Xu et al., 2013*). Another difference is that Eph receptor signalling clusters can exceed the size of the ligand contact zone several fold (*Wimmer-Kleikamp et al., 2004*), while we observed that DDR1 signalling clusters were strictly confined to the ligand contact area.

Interestingly, signalling between DDR1 dimers did not require identical cytoplasmic regions and was also observed with DDR2 as the donor kinase. These findings have functional implications in that catalytically incompetent DDR1 isoforms (*Alves et al., 2001*) could contribute to signalling when co-expressed with catalytically competent DDR1 isoforms or DDR2, with the signalling outcome likely dependent on the balance between catalytically active and inactive receptors, as has been suggested for Eph receptor signalling (*Truitt and Freywald, 2011*).

Recent studies have shown that oligomerisation beyond dimerisation also plays a role in signalling of RTKs that have traditionally been assumed to signal exclusively as dimers. Thus, EGF binding to the EGF receptor can result in formation of tetramers and higher-order oligomers, which may boost autophosphorylation of C-terminal tail tyrosines (*Clayton et al., 2005*; *Huang et al., 2016*; *Needham et al., 2016*). However, conflicting structural models have been proposed for the architecture and stoichiometry of EGF-bound EGF receptor multimers: one model involves self-association of ligand-bound dimers (*Huang et al., 2016*), while another model involves association via unoccupied ligand-binding sites, resulting in oligomers that can bind two EGF molecules at most (*Needham et al., 2016*). Our microscopic and biochemical data define a ligand-induced clustering mechanism and phosphorylation between neighbouring DDR1 dimers in trans; the size and structure of DDR1 signalling clusters and whether other molecules are required to co-associate with DDR1 remain to be determined.

# Materials and methods

## Cell culture

Human embryonic kidney (HEK) 293 cells, monkey Cos-7 fibroblast-like kidney cells, and human breast cancer MCF-7 cells were from ATCC, Manassas, VA, and were cultured in Dulbecco's modified Eagle's medium/F12 nutrient mixture (DMEM/F12; Invitrogen) supplemented with 2 mM L-glutamine, 100 units/ml penicillin, 100 μg/ml streptomycin and 10% foetal bovine serum. Human breast cancer T47D cells were obtained from the Cell Production Laboratory, Imperial Cancer Research Fund, UK, and were cultured in Roswell Park Memorial Institute medium (Invitrogen) supplemented with 2 mM L-glutamine, 100 units/ml penicillin, 100 μg/ml streptomycin and 10% foetal bovine serum. MCF-7 cells (on the list of commonly misidentified cell lines established by the International Cell Line Authentication Committee) and T47D cells were only used as example cell lines with endogenous DDR1 expression, for comparison of their DDR1 surface levels to those in our HEK-293 cell expression system; their authenticity was not confirmed. Neonatal normal human epidermal

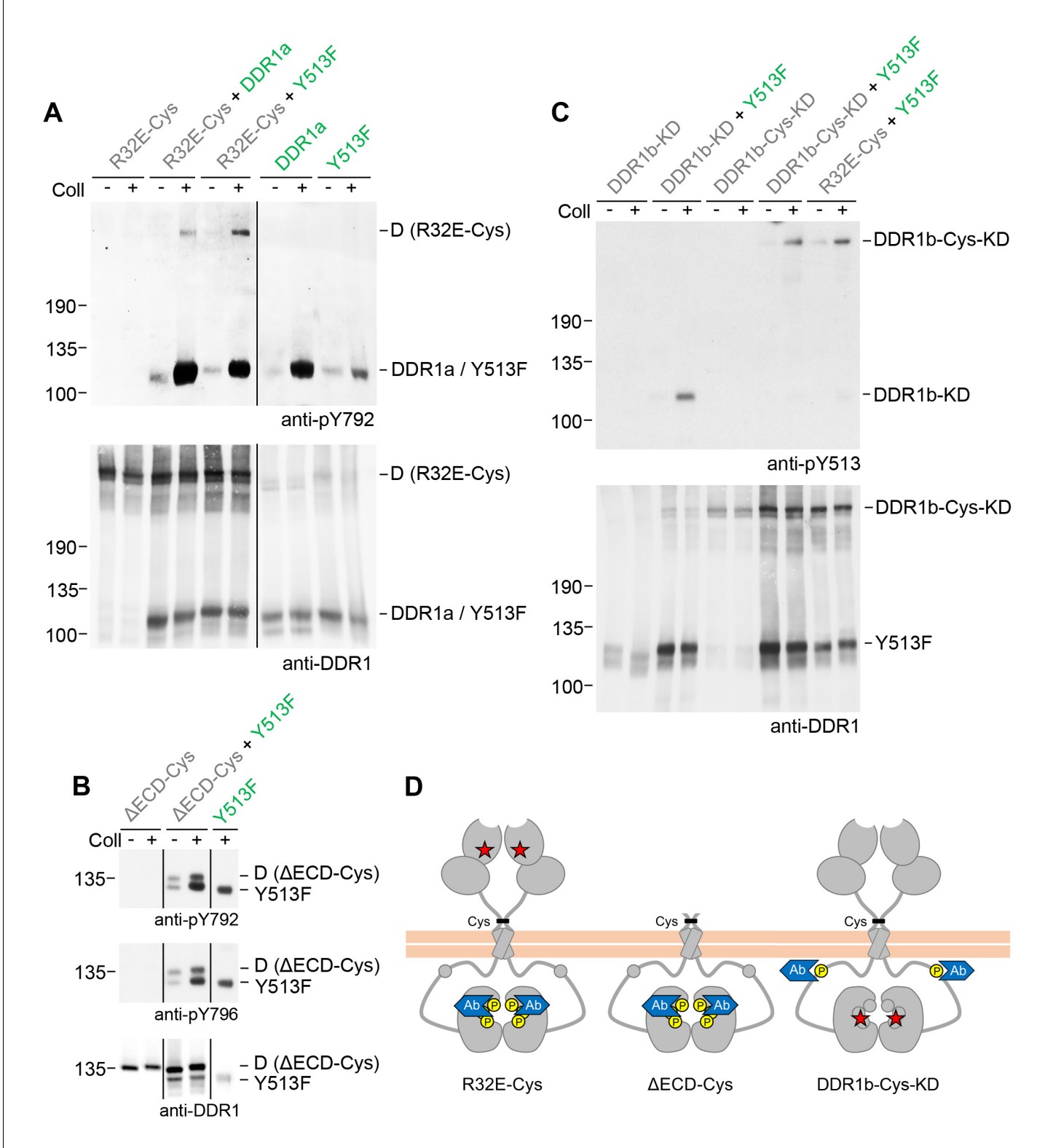

**Figure 6.** Signalling between dimers leads to activation loop phosphorylation of the receiver kinase but does not require receiver kinase activity. (A, B) Signalling-defective (DDR1b-R32E-Cys) or ectodomain-deletion (DDR1b-ΔECD-Cys) constructs were transiently expressed in HEK293 cells, either alone or in combination with DDR1a or DDR1b-Y513F, as indicated. (C) Kinase-inactive DDR1 constructs (DDR1b-KD or DDR1b-Cys-KD) were expressed in HEK293 cells, either alone or in combination with DDR1b-Y513F, as indicated. The cells were stimulated with collagen I (Coll) for 90 min at 37°C or left untreated. Cells were lysed in the presence of NEM, and aliquots of cell lysates were analysed by non-reducing SDS-PAGE followed by Western

*Figure 6 continued on next page*

*Figure 6 continued*

blotting. The blots were probed with phospho-specific Abs against activation loop tyrosine-792 or tyrosine-796 (**A, B**) or juxtamembrane tyrosine-513 (**C**), as indicated (upper blots), and re-probed with anti-DDR1 (lower blots). The positions of molecular mass markers are indicated on the left (in KDa). The positions of cysteine-linked dimeric (**D**) DDR1 receiver constructs, as well as DDR1a or DDR1b-Y513F donor DDR1 are indicated on the right. Note that in (**A**), the latter position includes heterodimers between DDR1b-R32E-Cys and DDR1a/DDR1-Y513F. (**D**) Schematic diagram showing collagen-induced activation loop (pY792) phosphorylation of DDR1b-R32E-Cys or DDR1b-ΔECD-Cys receiver constructs when co-expressed with donor DDR1, as well as Y513 phosphorylation of DDR1b-Cys-KD when co-expressed with DDR1b-Y513F. The blue shape denotes anti-pY792 or anti-pY513 used for Western blotting.

The following figure supplements are available for figure 6:

**Figure supplement 1.** Signalling between dimers leads to activation loop phosphorylation of the receiver kinase.

**Figure supplement 2.** Signalling between dimers leads to activation loop phosphorylation of the receiver kinase independently of receiver kinase activity.

keratinocytes (Lonza, Switzerland; passages 3–5) were cultured on a mitomycin C-treated feeder layer of 3T3 fibroblasts in DMEM/F12 medium supplemented with 1.8 mM $CaCl_2$, 100 units/ml penicillin, 100 μg/ml streptomycin, 10% foetal calf serum, 5 mM L-glutamine, 5 μg/ml insulin, 0.5 μg/ml hydrocortisone, 0.1 nM cholera toxin, and 10 ng/ml epidermal growth factor. All cells were grown at 37°C, 5% CO2. Cells were tested for mycoplasma contamination every four months and were confirmed negative.

## Chemicals and reagents

Bovine serum albumin (BSA) was obtained from Fisher Scientific (Loughborough, UK). Collagen I (acid-soluble from rat tail; C-7661) and collagen IV (acid-soluble from placenta; C-5533) were purchased from Sigma (Gillingham, UK). The collagen-mimetic DDR-selective peptide, (GPP)5-GPRGQOGVNleGFO-(GPP)5GPC-NH2, was obtained from Prof. Richard Farndale, University of Cambridge, UK. The peptide was synthesized by Fmoc (*N*-(9-fluorenyl)methoxycarbonyl) chemistry as a C-terminal amide on TentaGel R RAM resin in an Applied Biosystems Pioneer automated synthesizer and purified as described (*Raynal et al., 2006*). N-ethylmaleimide (NEM) was purchased from Sigma (E-1271).

The following primary Abs were used: rabbit anti-DDR1 (SC-532, Santa Cruz, Dallas, TX); goat anti-DDR1 extracellular domain Ab (AF2396, R&D Systems, Minneapolis, MN); goat anti-DDR2 (AF2538 from R&D Systems); rabbit anti-phospho-DDR1 (Tyr513) and rabbit anti-phospho-DDR1 (Tyr792) from Cell Signalling (Danvers, MA); rabbit anti-phospho-DDR2 (Tyr740) from R&D Systems; mouse anti-Flag IgG1 clone M2 (Sigma); rabbit anti-Flag (Sigma); and mouse anti-tubulin (Sigma). Anti-phospho-DDR1 (Tyr513), here referred to as anti-pY513, is a phospho-specific Ab that recognises phosphorylated tyrosine-513 in DDR1b. Anti-phospho-DDR1 (Tyr792), here referred to as anti-pY792, recognises phosphorylated tyrosine-792 in the activation loop of DDR1a and DDR1b. Anti-phospho-DDR2 (Tyr740), here referred to as anti-pY796, recognises phosphorylated Tyr740 in the activation loop of DDR2 and cross-reacts with the equivalent residue in DDR1, phosphorylated tyrosine-796. Mouse anti-DDR1 mAbs, 7A9 and IF10, were generated in our lab (*Carafoli et al., 2012*). The multimeric anti-DDR1 IgM Ab (mAb513) was used in the form of ascites and was a gift from Dr Michel Faure, SUGEN Inc. Secondary Abs were as follows: goat anti-rabbit Ig-horseradish peroxidase conjugated (P0448, DAKO A/S, Denmark); rabbit anti-goat Ig-horseradish peroxidase (Zymed Laboratories Inc., San Francisco, CA); sheep anti-mouse Ig-horseradish peroxidase (Amersham Biosciences, Little Chalfont, UK); goat anti-mouse IgG FITC-conjugated (F-9006, Sigma); goat anti-mouse IgG Alexa-Flour-488 (Invitrogen); goat anti-mouse IgG1 Alexa-Fluor-488 (Invitrogen); goat anti-rabbit Alexa-Fluor-488 (Invitrogen), goat anti-mouse IgG2b Alexa-Fluor-555 (Invitrogen); goat anti-rabbit IgG Alexa-Fluor-555 (Invitrogen); and goat anti-rabbit IgG Alexa-Fluor-647 (Invitrogen).

## DNA constructs and site directed mutagenesis

DDR1b-R32E and DDR1b-L152E were previously generated (*Carafoli et al., 2012*). DDR1b-KD (K655A) and DDR1a-KD (K655A) constructs were generated by strand overlap extension PCR as

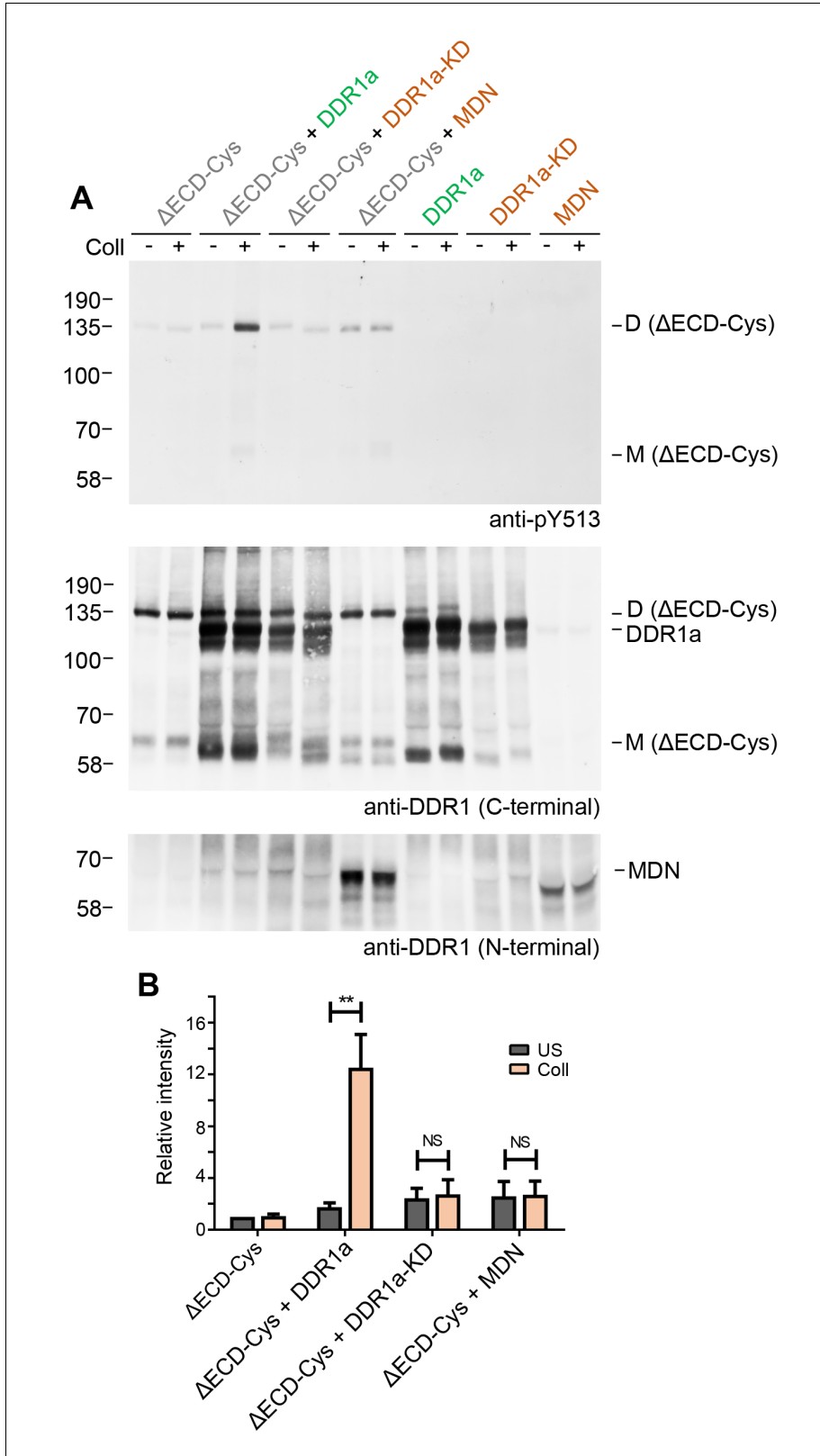

**Figure 7.** Signalling between dimers requires kinase activity of the donor kinase. (**A**) The ectodomain-deletion construct DDR1b-*ΔECD-Cys* was transiently expressed in HEK293 cells, either alone or co-expressed with DDR1a, kinase-inactive DDR1a (DDR1a-KD) or with a truncated DDR1 construct lacking the kinase domain (DDR1-MDN), as indicated. The cells were stimulated with collagen I (Coll) for 90 min at 37°C or left untreated. Cells were lysed in

*Figure 7 continued on next page*

*Figure 7 continued*

the presence of NEM, and aliquots of cell lysates were analysed by non-reducing SDS-PAGE followed by Western blotting. The blot was probed with phospho-specific anti-pY513 (upper blot), and re-probed with anti-DDR1 (middle blot, C-terminal Ab). Expression of DDR1-MDN was verified by blotting with anti-DDR1 Abs against the ectodomain (lowest blot, N-terminal Ab). The positions of molecular mass markers are indicated on the left (in KDa). The positions of cysteine-linked dimeric DDR1b-ΔECD-Cys (D) and of mature monomeric DDR1b-ΔECD (M) receiver constructs, as well as of DDR1a or DDR1-MDN donor DDR1 are indicated on the right. (**B**) Quantitation of receiver DDR1 pY513 signals, normalised to respective DDR1 signals, expressed as relative band intensity with respect to the lowest signals on the blots (unstimulated DDR1b-ΔECD-Cys). US, unstimulated; Coll, stimulation with collagen I. **p<0.01; NS, no significance; n = 5.

described previously (*Leitinger, 2003*). DDR1b-W53A and DDR1b-Y513F constructs were generated with the QuickChange method using wild type DDR1b cDNA cloned into a pSP72 vector (Promega) as a template. All cysteine substitution mutants have the T416C mutation and were generated by subcloning using a previously made cDNA construct encoding DDR1b-T416C. To obtain DDR1b-W53A-Cys, DDR1b-R32E-Cys and DDR1b-L152E-Cys, DDR1b-W53A, DDR1b-R32E and DDR1b-L152E constructs were cut with EcoRI and SacI and subcloned into a pGEM-3Z (Promega)-based vector encoding DDR1b-T416C (*Xu et al., 2014*), after the corresponding wild-type EcoRI-SacI sequence was removed. The DDR1b-KD-Cys construct was generated by subcloning the XhoI-BamHI fragment of DDR1b-KD into a pRK5 (BD PharMingen)-based vector encoding DDR1b-T416C, after the corresponding wild-type sequence was removed. DDR1b-ΔECD (with an N-terminal Flag tag followed by a short linker) was generated with the FastCloning PCR method (*Li et al., 2011*), eliminating residues Gly24 to Lys410 of WT-DDR1b. The mature construct after signal peptide cleavage has the following N-terminal sequence: DADMK[23]DYKDDDKSGSAEGSPTA[417]ILI (Flag tag and SGS linker underlined; first three transmembrane domain residues in bold). An N-terminally Flag-tagged DDR1b cDNA, cloned into pGEM-3Z vector, was used as a template. DDR1b-ΔECD-Cys was generated by introducing the T416C mutation using the QuickChange method. DDR1b-ΔECD-TM1 and DDR1b-ΔECD-Cys-TM1 were generated by introducing the L430G/L431P mutation (*Noordeen et al., 2006*) using DDR1b-ΔECD or DDR1b-ΔECD-Cys constructs as templates. The cytoplasmic DDR1 deletion construct MDN (DDR1b truncated after Arg525) was a gift from Dr Michel Faure, SUGEN Inc.

All cDNAs encoding mutant DDR1b were cloned into the mammalian expression vector pRK5 (BD Pharmingen) for transient transfection in cells. Restriction and modification enzymes were purchased from New England Biolabs (Hitchin, UK) or Promega (Southampton, UK). All PCR-derived DNA constructs were verified by DNA sequencing. PCR primers used for generating the mutant constructs can be obtained on request.

## Transfection of cells

HEK293 cells were grown in 24-well (for DDR1 autophosphorylation assays) or 6-well (for flow cytometry and comparison of protein expression levels) tissue culture plates and transfected with relevant DDR1 expression plasmids by calcium phosphate precipitation. Cos-7 cells were grown on coverslips in 24-well tissue culture plates and transfected with Fugene HD (Promega, Madison, WI) according to the manufacturer's instructions.

## DDR1 autophosphorylation assay

The assay was performed as described before (*Leitinger, 2003*). Briefly, 24 hr after transfection, the cells were incubated with serum-free medium for 16 hr. Cells were then stimulated with 10 µg/ml collagen I, 50 µg/ml collagen IV, 50 µg/ml collagen-mimetic DDR-selective peptide, or the multimeric anti-DDR1 Ab (mAb513, 1:500 dilution from ascites) for 90 min at 37°C. Cells were lysed in 1% Nonidet P-40, 150 mM NaCl, 50 mM Tris, pH 7.4, 1 mM EDTA, 1 mM phenylmethylsulfonyl fluoride, 50 µg/ml aprotinin, 1 mM sodium orthovanadate, and 5 mM NaF. Aliquots of the lysates were analysed by reducing SDS-PAGE on 7.5% polyacrylamide gels. In experiments with cysteine mutants, the lysis procedure was as described (*Xu et al., 2014*). Cells were lysed in lysis buffer containing 30 mM NEM to avoid potential non-native disulphide formation during lysis, and aliquots of lysates

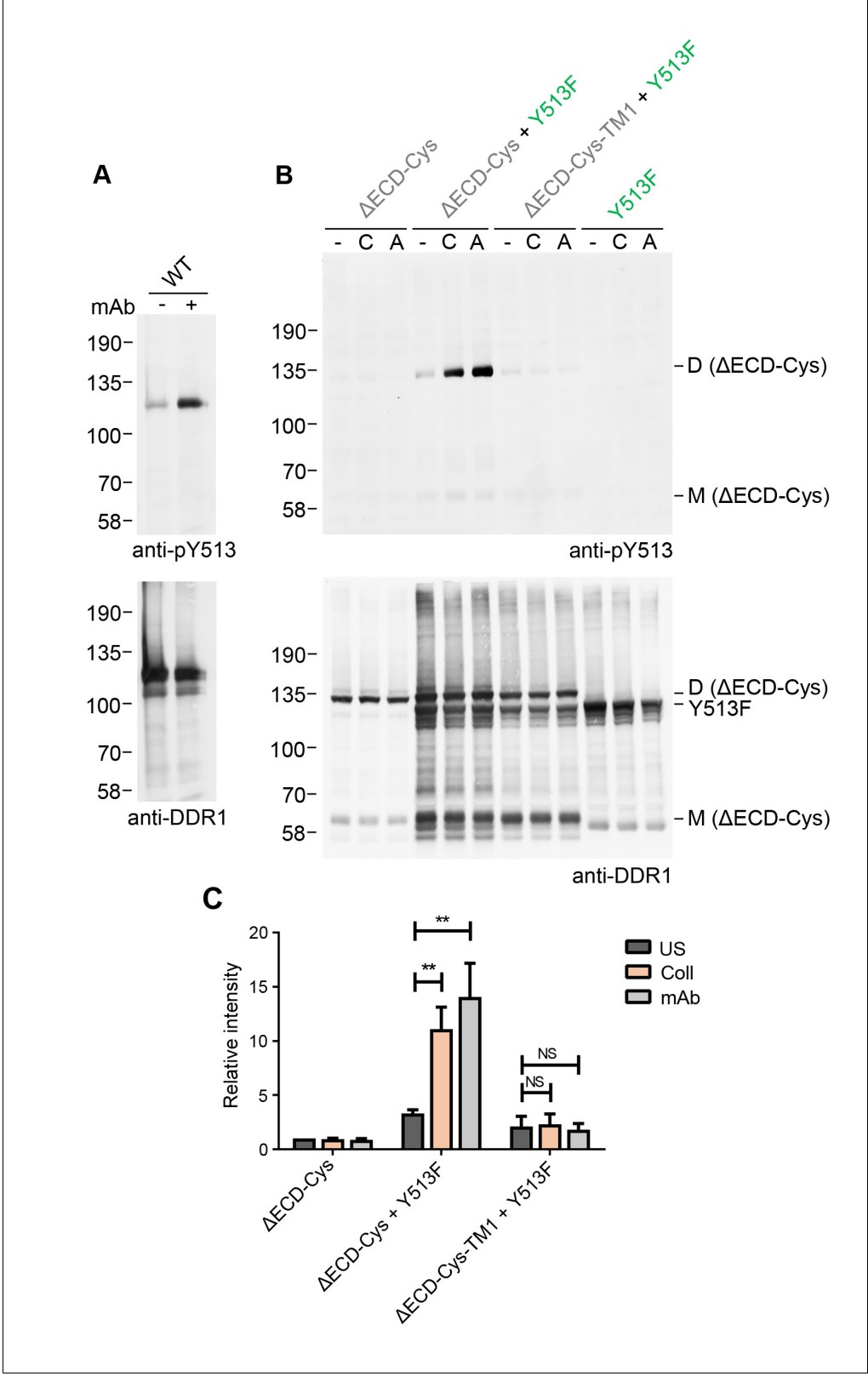

**Figure 8.** Collagen-independent activation by a multimeric anti-DDR1 Ab leads to signalling between dimers. (**A**) Wild-type DDR1b was transiently expressed in HEK293 cells, and the cells were incubated with mAb513 for 90 min at 37°C or left untreated. Cell lysates were analysed by reducing SDS-PAGE followed by Western blotting. (**B**) The ectodomain-deletion construct DDR1b-ΔECD-Cys was transiently expressed in HEK293 cells, either alone or co-

*Figure 8 continued on next page*

*Figure 8 continued*

expressed with DDR1b-Y513F, as indicated. The cells were stimulated with collagen I (C) or mAb513 (A) for 90 min at 37°C or left untreated. Cells were lysed in the presence of NEM, and aliquots of cell lysates were analysed by non-reducing SDS-PAGE followed by Western blotting. The blots were probed with phospho-specific anti-pY513 (upper blots), and re-probed with anti-DDR1 (lower blots). The positions of molecular mass markers are indicated on the left (in KDa). The positions of cysteine-linked dimeric DDR1b-ΔECD-Cys (D) and mature monomeric DDR1b-ΔECD (M) receiver constructs, as well as DDR1b-Y513F donor DDR1 are indicated on the right. (**C**) Quantitation of pY513 signals, normalised to respective DDR1 signals, expressed as relative band intensity with respect to the lowest signals on the blots (unstimulated DDR1b-ΔECD-Cys). US, unstimulated; Coll, stimulation with collagen I; mAb, stimulation with mAb513. **p<0.01; NS, no significance; n = 4.

were analysed by non-reducing SDS-PAGE on 5% or 7.5% polyacrylamide gels. This was followed by blotting onto nitrocellulose membranes. The blots were probed first with anti-phospho-DDR1 Abs followed by horseradish peroxidase conjugated goat anti-rabbit secondary Abs, then stripped in Antibody Stripping Solution (Alpha Diagnostic International, San Antonio, Texas) and re-probed with anti-DDR1 Abs followed by horseradish peroxidase conjugated goat anti-rabbit secondary Abs. To detect expression of DDR2 or the truncated DDR1 construct lacking the kinase domain (DDR1-MDN) in co-expression experiments, blots were probed first with anti-DDR2 or anti-DDR1 against the ecto-domain, respectively, followed by horseradish peroxidase conjugated anti-goat secondary Abs. Signal detection was performed using Enhanced Chemiluminescence Plus or Enhanced Chemiluminescence Select reagents (Amersham Biosciences) on an Ettan DIGE imager, a Typhoon FLA 9500 Imager (GE Healthcare Biosciences) or X-ray film.

## DDR1 protein expression

HEK293, T47D and MCF7 cells were grown in 6-well tissue culture plates and primary keratinocytes were grown in 10 cm dishes. HEK293 cells were transfected with varying amounts of DDR1b expression plasmid. Cells were serum-starved for 16 hr and lysed as above. Equal amounts of protein were analysed by reducing SDS-PAGE on 7.5% polyacrylamide gels. This was followed by blotting onto nitrocellulose membranes. The blots were probed with anti-DDR1 Ab followed by horseradish peroxidase conjugated goat-anti-rabbit secondary Ab, or anti-tubulin Ab followed by horseradish peroxidase conjugated sheep-anti-mouse secondary Ab. Signal detection was performed as above.

## Cell surface immunofluorescence staining

Cos-7 cells transiently expressing the ectodomain deletion constructs, DDR1b-ΔECD, DDR1b-ΔECD-TM1, DDR1b-ΔECD-Cys or DDR1b-ΔECD-Cys-TM1, which contain N-terminal Flag tags, were washed with PBS, and incubated with mouse anti-Flag IgG1 for 1 hr on ice to label cell surface DDR1 constructs. This was followed by washing, fixing with 4% paraformaldehyde (Sigma) in PBS for 15 min at room temperature, blocking in 8% BSA in PBS for 1 hr, and incubation with anti-mouse IgG1 Alexa-Fluor-488 secondary Ab. Cos-7 cells transiently expressing DDR1b wildtype were stimulated with 10 μg/ml collagen I or a mixture of 10 μg/ml collagen and 10 μg/ml mouse anti-DDR1 7A9 IgG1 mAb, for 10 min at 37°C. The cells were washed with PBS, and incubated with mAb 7A9 for 1 hr on ice to label cell surface DDR1. This was followed by fixation and incubation with secondary Ab as above. The coverslips were mounted on slides with Prolong Gold Antifade Reagent (Invitrogen). The cells were examined using an Olympus BX-51 widefield microscope.

## Flow cytometry

For detection of cell surface expression of DDR1 ectodomain deletion mutants, HEK293 cells in 6-well tissue culture plates were transfected with N-terminally Flag-tagged DDR1 ectodomain deletion expression plasmids (DDR1b-ΔECD, DDR1b-ΔECD-TM1, DDR1b-ΔECD-Cys or DDR1b-ΔECD-Cys-TM1) and grown for 48 hr before being dissociated and resuspended in PBS containing 1% BSA. The cells were incubated for 30 min on ice with rabbit anti-Flag Ab in 100 μl of PBS/BSA. Cells were then washed three times with PBS/BSA and incubated with goat anti-rabbit Alexa-Fluor-488 Ab for 30 min on ice. After three washes as above, the cells were resuspended in 2% formaldehyde in PBS. For comparison of surface DDR1 expression, HEK293, T47D and MCF7 cells were grown in 6-well

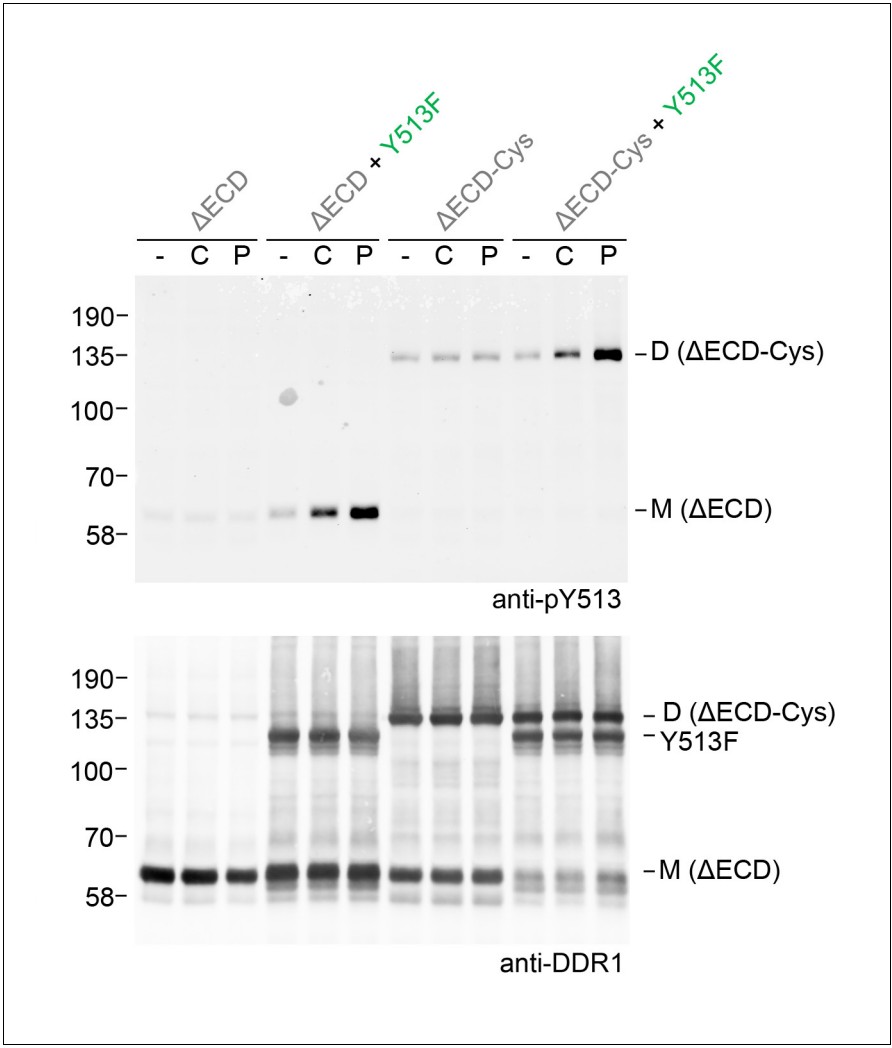

**Figure 9.** Ligand multivalency is not required for signalling between dimers. The ectodomain-deletion constructs DDR1b-ΔECD or DDR1b-ΔECD-Cys were transiently expressed in HEK293 cells, either alone or co-expressed with DDR1b-Y513F, as indicated. The cells were stimulated with collagen I (C) or a collagen-mimetic DDR-selective peptide (P) for 90 min at 37°C or left untreated. Cells were lysed in the presence of NEM, and aliquots of cell lysates were analysed by non-reducing SDS-PAGE followed by Western blotting. The blot was probed with phospho-specific anti-pY513 (upper blot), and re-probed with anti-DDR1 (lower blot). The positions of molecular mass markers are indicated on the left (in KDa). The positions of cysteine-linked dimeric DDR1b-ΔECD-Cys (D) and mature monomeric DDR1b-ΔECD (M) receiver constructs, as well as DDR1b-Y513F donor DDR1 are indicated on the right.

tissue culture plates and primary keratinocytes were grown in 10 cm dishes. HEK293 cells were transfected with varying amounts of DDR1b expression plasmid. Cells dissociation, Ab staining and fixation were done as above, except that the primary Ab was anti-mouse DDR1 (mAb 7A9) and the secondary Ab was FITC-conjugated goat-anti-mouse IgG. Data were collected on a BD LSRFortessa cell analyser using BD FACSDiva software 6.0 (BD Biosciences) and further analysed on FlowJo software 10.2 (Tree Star, Inc.).

## Coating of beads with collagen

3 µm diameter latex beads (Sigma) were washed in distilled water, followed by washes in coating buffer (50 mM Tris, 100 mM NaCl, pH 8.5) and incubation with 20 µg/ml collagen I in coating buffer overnight at 4°C on a rotating wheel. Control beads were coated with 20 µg/ml BSA. The beads

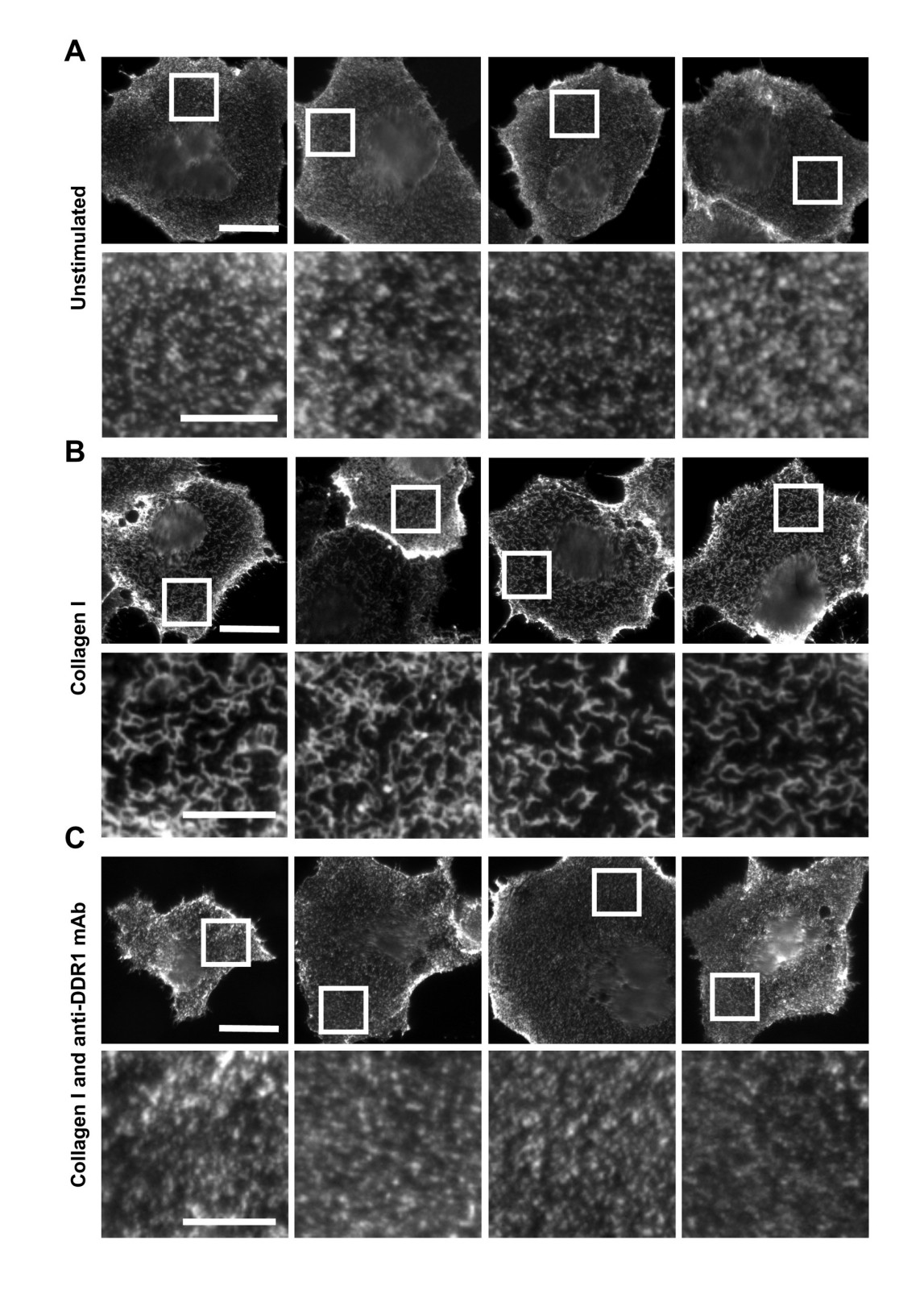

**Figure 10.** Function-blocking anti-DDR1 mAb inhibits collagen-induced DDR1 clustering. Wild-type DDR1b was transiently expressed in Cos-7 cells and either left unstimulated (**A**), stimulated with 10 μg/ml collagen I (**B**), or stimulated with 10 μg/ml collagen I in the presence of a function-blocking anti-DDR1 mAb at 10 μg/ml (**C**), for 10 min at 37°C. Cells from all conditions were incubated on ice with a mouse monoclonal Ab against the DDR1

*Figure 10 continued*

ectodomain, before fixation and incubation with anti-mouse Alexa-Fluor-488 Ab, and mounting. Cells were imaged using a widefield microscope. White boxes in top rows indicate corresponding areas shown at higher magnification in lower rows. Scale bars, 30 μm (top rows) and 10 μm (bottom rows).

were then washed in PBS, blocked in 1% BSA in PBS for 1 hr, and re-suspended in serum-free medium for use on cells.

## Bead recruitment assay

Cos-7 cells transiently expressing wild-type DDR1b were grown on coverslips and incubated with serum-free medium for 16 hr. Cells were then stimulated with collagen-coated beads or control BSA-coated beads for the indicated time at 37°C. After stimulation, cells were washed with PBS, and incubated with mouse anti-DDR1 7A9 IgG1 mAb for 1 hr on ice to label surface DDR1. This was followed by washing in PBS, fixing with 4% paraformaldehyde in PBS for 15 min at room temperature, washing in PBS, permeabilisation with 0.1% Triton X-100 (Sigma) in PBS for 5 min, blocking in 8% BSA in PBS for 1 hr, and incubation with rabbit anti-pY513 Ab. After washing, the cells were then incubated with anti-mouse IgG Alexa-Fluor-488 and anti-rabbit Alexa-Fluor-555 secondary Abs. Cos-7 cells transiently expressing the ectodomain deletion construct DDR1b-ΔECD (which contains an N-terminal Flag tag), DDR1b-Y513F or both were stimulated with collagen-coated beads for 2 hr at 37°C. After stimulation, cells were washed with PBS, and incubated with mouse anti-Flag IgG1 and mouse anti-DDR1 ectodomain IF10 IgG2b Abs for 1 hr on ice to label surface DDR1b-ΔECD and DDR1b-Y513F, respectively. This was followed by washing, fixing, permeabilisation, blocking, and incubation with rabbit anti-pY513 Ab as described above. The cells were then incubated with anti-mouse IgG1 Alexa-Fluor-488, anti-mouse IgG2b Alexa-Fluor-555 and anti-rabbit Alexa-Fluor-647 secondary Abs. The coverslips were mounted on slides with Prolong Gold Antifade Reagent (Invitrogen). The cells were examined using an Olympus BX-51 widefield microscope.

## Statistical analysis

All experiments were performed at least three times with similar results. Blots show representative examples. Intensities of protein bands were quantitated using ImageJ software. All data are expressed as mean ± standard error of the mean (SEM) from at least three independent experiments. Statistical significance was tested with a Kruskal-Wallis multiple comparison test, followed by Mann-Whitney tests using GraphPad Prism 7 (San Diego, CA). Statistical significance was set at a p value < 0.05.

## Acknowledgements

We thank Dominique Bihan and Richard Farndale (University of Cambridge) for supplying the collagen-mimetic peptide and Erhard Hohenester for critical reading of the manuscript.

## Additional information

### Funding

| Funder | Grant reference number | Author |
|---|---|---|
| Medical Research Council | MRC Doctoral Training Partnership PhD Studentship | Victoria Juskaite |
| National Heart and Lung Institute | NHLI Foundation studentship | David S Corcoran |

The funders had no role in study design, data collection and interpretation, or the decision to submit the work for publication.

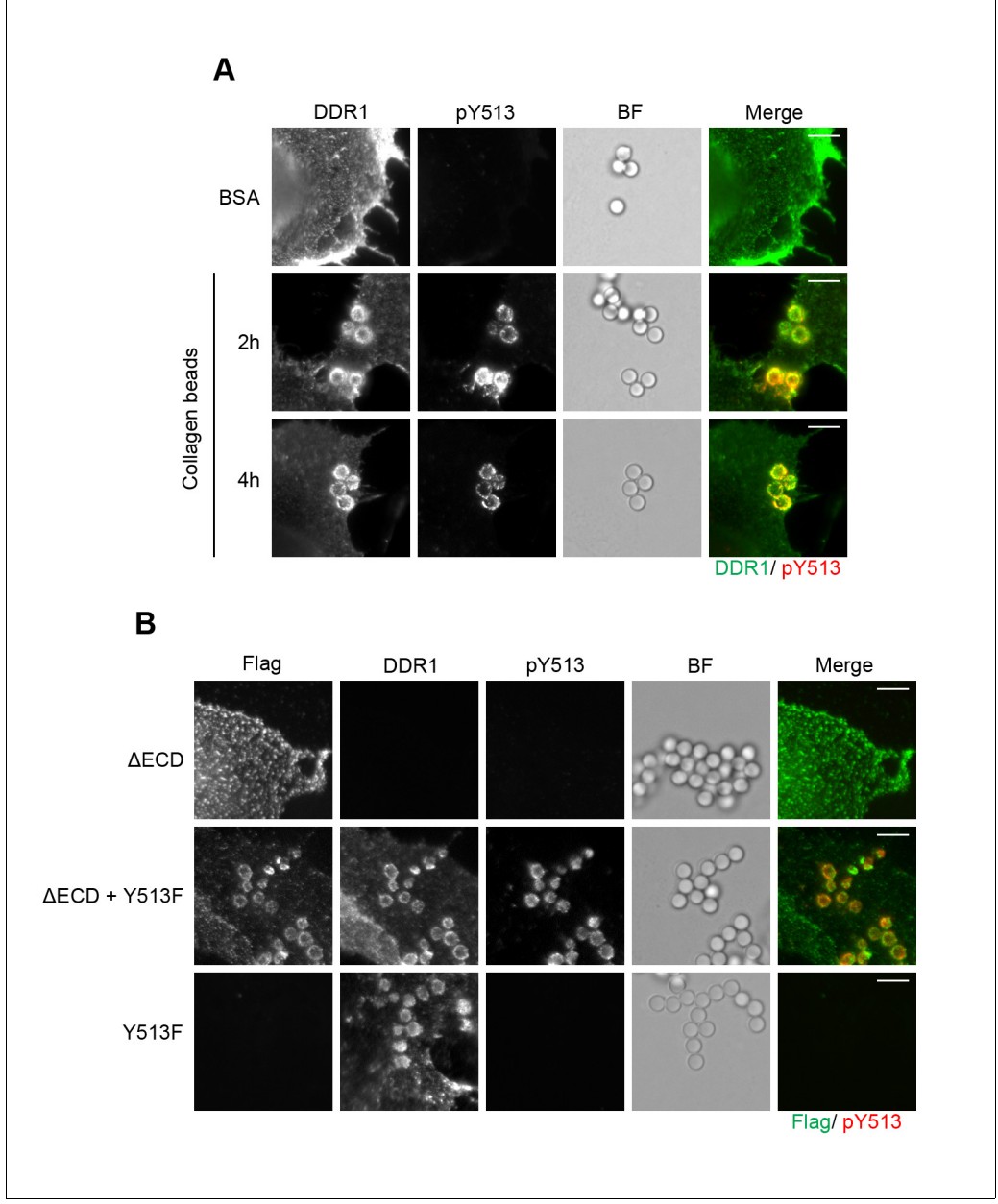

**Figure 11.** Collagen induces local activation of DDR1 and leads to recruitment of ectodomain-deletion DDR1 into the ligand-binding contact zone. (**A**) Wild-type DDR1b was transiently expressed in Cos-7 cells and stimulated with collagen-coated latex beads (3 μm diameter) for 2 or 4 hr at 37°C. Control beads were coated with BSA and incubated for 2 hr at 37°C. Cells were incubated on ice with a mouse monoclonal Ab against the DDR1 ectodomain, before fixation and permeabilisation and incubation with rabbit anti-pY513 Ab. Cells were then incubated with anti-mouse Alexa-Fluor-488 and anti-rabbit Alexa-Fluor-555 secondary Abs. DDR1 (green) and pY513 (red) staining are shown in the merge image (right panel). (**B**) The ectodomain deletion construct DDR1b-ΔECD (which contains an N-terminal Flag tag) and DDR1b-Y513F were expressed either alone or in combination in Cos-7 cells. Cells were stimulated with collagen-coated latex beads (3 μm diameter) for 2 hr at 37°C. Cells were incubated on ice with mouse anti-Flag IgG1 and mouse anti-DDR1 ectodomain IgG2b Abs, followed by fixation and permeabilisation and incubation with rabbit anti-pY513 Ab. Cells were then incubated with anti-mouse IgG1 Alexa-Fluor-488, anti-mouse IgG2b Alexa-Fluor-555, and anti-rabbit Alexa-Fluor-647 secondary Abs. Flag (green) and pY513 (red) staining are shown in the merge image (right panel). BF, Brightfield. Scale bars, 10 μm.

## Author contributions

VJ, Writing—original draft, Acquisition of data; Analysis and interpretation of data; Experimental design; Performed all experiments except those for Figure 10; DSC, Acquisition of data; Analysis and interpretation of data; Performed experiments for Figure 10; Assisted with revised draft; BL, Conceptualization, Supervision, Writing—original draft, Conception and design of the study; Analysis and interpretation of data

## Author ORCIDs

Birgit Leitinger, http://orcid.org/0000-0003-2426-1179

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
