## [Decision Letter]

Thank you for submitting your article "Collagen induces activation of DDR1 through lateral dimer association and phosphorylation between dimers" for consideration by *eLife*. Your article has been favorably evaluated by Fiona Watt (Senior Editor) and three reviewers, one of whom, Reinhard Fässler (Reviewer #1), is a member of our Board of Reviewing Editors.

The reviewers have discussed the reviews with one another and the Reviewing Editor has drafted this decision to help you prepare a revised submission.

Summary:

The findings of your paper show that (1) ligand binding-induced clustering leads to the activation of DDR dimers, (2) active DDR dimers phosphorylate neighbouring DDR dimers in the juxtamembrane and activation loop, (3) the ectodomain is not required for these phosphorylation events in the receiving neighbouring DDR dimers, and (4) the TM domain interaction is essential for donor as well as neighbouring receiver DDR dimers. Despite some clear strengths of the submitted manuscript, the research has some important shortcomings and limitations, which the authors should address in a revised submission:

Essential revisions:

1) It is necessary to exclude that over-expressed DDR does not lead to artificial aggregates, How do the overexpressed surface levels compare to wild type level? Key experiments could be repeated at low expression levels if overexpressed levels are indeed much higher than endogenous levels.

2) It is necessary to show functional relevance (e.g. adhesion) of DDR clustering.

3) The dimer size is unclear. Confirm the dimer size in SDS-PAGE with appropriate molecular size markers. Indicate molecular weights in gels and blots.

4) The authors use cysteine mutants to further separate phosphorylation within dimers and across different dimers. It is shown that for inter-dimer phosphorylation the receiver requires specific transmembrane sequences and the donor requires catalytic activity. Although activation loop phosphorylation can also occur across dimers, it appears that intra-dimer phosphorylation of the activation loop is more efficient (Figure 6 and Figure 6). This indicates that important ligand induced signaling also occurs independent of receptor clustering. Whereas activation loop phosphorylation has a clear role in DDR kinase activation, the role for juxtamembrane phosphorylation is less clear. Therefore, although the data does quite convincingly show that juxtamembrane phosphorylation does occur across dimers, its importance is not clear. It is necessary to clarify whether the phosphorylation of the activation loop is due to cross-dimer and/or intra-dimer phosphorylation.

[Editors' note: further revisions were requested prior to acceptance, as described below.]

Thank you for resubmitting your work entitled "Collagen induces activation of DDR1 through lateral dimer association and phosphorylation between dimers" for further consideration at *eLife*. Your revised article has been favorably evaluated by Fiona Watt (Senior Editor) and three reviewers, one of whom is a member of our Board of Reviewing Editors.

The manuscript has been improved but there are some remaining issues that need to be addressed before acceptance.

Your figure provided to the reviewers demonstrates that DDR function prevents formation of larger aggregates. This figure should be included into the manuscript to underline the functional relevance of your findings.

We also believe that activation loop phosphorylation across dimers occurs. However, your data suggest that the more efficient mechanism occurs within dimers. The weak dimer phosphorylation exclusively occurs across dimers (inter-dimer), whereas the strong monomer phosphorylation is the sum of intra and inter dimer phosphorylation, suggesting that this sum is dominated by intra-dimer phosphorylation. There is no evidence that inter-dimer phosphorylation of the activation loop is the major event or relevant. As long as phosphorylation within dimers occurs at least as efficient as across dimers, lateral association and activation loop phosphorylation is not important.

Since the paper focuses on Y513 phosphorylation, it is not necessary to show that lateral association is responsible for activation loop phosphorylation. However, due to the two points above (lateral association is neither shown (1) to be activating nor (2) relevant for activation loop phosphorylation.) the statement 'Our data suggest that DDRs are activated by a mechanism in which collagen binding promotes lateral association of DDR1 dimers, which induces activation loop phosphorylation and thereby releases cis-autoinhibition of the kinase domain.' should be toned down and re-phrased appropriately.

---

## [Author Response]

*Essential revisions:*

*1) It is necessary to exclude that over-expressed DDR does not lead to artificial aggregates, How do the overexpressed surface levels compare to wild type level? Key experiments could be repeated at low expression levels if overexpressed levels are indeed much higher than endogenous levels.*

We agree with the reviewers’ concern and have repeated key experiments at low DDR1 expression levels that clearly demonstrate dimer phosphorylation of receiver kinases, similar to the results of our original experiments. The new data rule out that the phosphorylation between neighbouring DDR1 dimers is due to aggregation of DDR1 at artificially high expression levels.

DDR1 is expressed in a wide range of tissues but no systematic study has been carried out to compare expression levels of endogenous DDR1 in different cell types. Immortalised cancer cell lines have varying levels of surface DDR1. We used flow cytometry to compare surface levels of DDR1 in our HEK293 cell expression system with the levels on primary human keratinocytes and two commonly used breast cancer cell lines. The new data (Figure 3—figure supplement 1) show that the mean DDR1 surface levels resulting from the transfection conditions used in most of our experiments (Figure 2–Figure 9) are similar to those on T47D breast cancer cells (Figure 3—figure supplement 1, HEK-5ug DNA vs. T47D) and approximately 3 times higher than those on primary keratinocytes. We have repeated key co-expression experiments under conditions that result in DDR1 surface levels that are comparable with those on primary keratinocytes (Figure 3—figure supplement 1; HEK-1ug DNA vs Keratinocytes). Detection of the phosphorylation signal was possible by using the most sensitive ECL substrates that are commercially available. Figure 3—figure supplement 2 shows collagen-induced phosphorylation of DDR1-R32E-Cys and DDR1-ΔECD-Cys receiver dimers when co-expressed with DDR1-Y513F donor kinase, using a range of different expression levels.

Because our experimental approach relies on co-expression of mutant DDR1 constructs, our experiments can only be done under transfection conditions. Collagen-induced DDR1 phosphorylation is a characteristically slow process, which would be expected to occur with faster kinetics if over-expression lead to a different form, or enhanced, levels of DDR1 aggregation. However, this is not the case, further arguing against an activation mechanism that is dependent on very high levels of DDR1 expression.

*2) It is necessary to show functional relevance (e.g. adhesion) of DDR clustering.*

In a previous publication (Carafoli et al., Structure, 2012) we identified a number of function-blocking anti-DDR1 antibodies we believe are interfering with DDR1 clustering. Blocking DDR1 clustering by other means is likely to result in a similar complete inhibition of ligand-induced DDR1 phosphorylation, with functional consequences for any cellular process that relies on DDR1 kinase activity.

In unpublished experiments that go beyond the scope of the current manuscript, we have shown that collagen binding to DDR1 leads to re-distribution of DDR1 on the cell surface from a punctate into a more compact structure (see Figure 12). This phenomenon can be described as ligand-induced clustering. Because this happens with fast kinetics, before phosphorylation is detected, our current molecular model of DDR1 activation involves ligand-induced receptor clustering as the first step, followed by kinase activation and further down-stream signalling. As can be seen in Figure 12, collagen-induced clustering can be prevented by addition of our function-blocking anti-DDR1 monoclonal antibodies. These antibodies bind to an extracellular epitope and allosterically block collagen-induced DDR1 phosphorylation (Carafoli et al., Structure, 2012). Together with the data attached to this rebuttal, this demonstrates that clustering is a key step in DDR1 activation. We note that our anti-DDR1 antibodies also block DDR1-mediated affinity regulation of collagen-binding integrins, which leads to enhanced adhesion (Xu et al., PLOS one, 2012). Hence DDR1 clustering is functionally important for cell adhesion.

Author response image 1.Anti-DDR1 mAbs inhibit collagen-induced DDR1 clustering.Cos-7 cells expressing DDR1 were incubated with 10 μg/ml collagen I for 10 minutes at 37°C, left unstimulated, or incubated with collagen I in the presence of function-blocking anti-DDR1 mAbs. Cells were incubated on ice with a mouse monoclonal Ab against the DDR1 ectodomain, before fixation and incubation with anti-mouse Alexa-Fluor-488. Cells were imaged on a widefield microscope.**DOI:**
http://dx.doi.org/10.7554/eLife.25716.021

*3) The dimer size is unclear. Confirm the dimer size in SDS-PAGE with appropriate molecular size markers. Indicate molecular weights in gels and blots.*

In a previous publication on disulphide-linked DDR1 constructs (including DDR1-T416C used in the present study), we showed blots of non-reducing gels with molecular mass markers up to 460 kDa (Xu et al., JBC, 2014). From these blots, it can be seen that full-length DDR1-Cys dimers are ~240 kDa, compared with ~120 kDa for DDR1 monomers on the same blot. The data shown in the new Figures (Figure 3—figure supplement 2 and Figure 6—figure supplement 2) are shown with the same molecular size markers up to 460 kDa, as in our original study.

*4) The authors use cysteine mutants to further separate phosphorylation within dimers and across different dimers. It is shown that for inter-dimer phosphorylation the receiver requires specific transmembrane sequences and the donor requires catalytic activity. Although activation loop phosphorylation can also occur across dimers, it appears that intra-dimer phosphorylation of the activation loop is more efficient (Figure 6 and Figure 6). This indicates that important ligand induced signaling also occurs independent of receptor clustering. Whereas activation loop phosphorylation has a clear role in DDR kinase activation, the role for juxtamembrane phosphorylation is less clear. Therefore, although the data does quite convincingly show that juxtamembrane phosphorylation does occur across dimers, its importance is not clear. It is necessary to clarify whether the phosphorylation of the activation loop is due to cross-dimer and/or intra-dimer phosphorylation.*

There is indeed strong activation loop phosphorylation at the position of donor DDR1a or DDR1-Y513F (Figure 6), but we do not agree with the reviewers’ conclusion that this signal necessarily results from phosphorylation within a dimer. It is equally possible that activation loop phosphorylation occurs by the same inter-dimer mechanism that we unequivocally demonstrate for Y513.

With regards to the activation loop phosphorylation signals shown in Figure 6 and Figure 6—figure supplement 1, we note that the signal obtained at the position of DDR1a and DDR1-Y513F represents both homodimers of functional receptor (DDR1a or DDR1-Y513F) and any heterodimers that might form between DDR1-R32E-Cys and DDR1a or DDR1-R32E-Cys and DDR1-Y513F. Therefore, these signals will also include some DDR1-R32E-Cys molecules; we have now clarified in the figure legends to Figure 6 and Figure 6—figure supplement 1 that the position of DDR1a/Y513F includes heterodimers. Furthermore, fully functional receptor constructs (DDR1a, DDR1-Y513F) are expected to be phosphorylated very efficiently, since they functionally engage with the ligand, in contrast to the receiver kinase. Local signal propagation involving functional DDR1 may be more effective than recruiting signalling-incompetent DDR1 molecules into functional signalling clusters, which could account for the difference in autophosphorylation signals seen in Figure 6. We further note that, in our hands, DDR1a activation loop phosphorylation signals are usually stronger than those of DDR1b (e.g. Figure 6 and Figure 6—figure supplement 1, right hand side), for reasons that are unclear at present. This may account for the particularly strong activation loop signals at the DDR1a position when DDR1-R32E-Cys is co-expressed with DDR1a.

The reviewers would like us to clarify whether activation loop phosphorylation is due to inter-dimer phosphorylation or phosphorylation within a dimer. The experiments that we present in Figure 6 and Figure 6—figure supplement 1 clearly show that activation loop phosphorylation of the receiver can occur as a consequence of inter-dimer phosphorylation. We have now extended this point and provide evidence that a kinase-dead receiver dimer is also phosphorylated on the activation loop (new Figure; Figure 6—figure supplement 2). These data clearly demonstrate that activation loop phosphorylation of receiver kinase occurs as a result of phosphorylation in trans, in the absence of receiver kinase activity. Because our experimental setup can only detect receiver (and not donor) kinases in dimeric form, it is not possible to test whether donor kinase phosphorylation occurs as a result of inter-dimer phosphorylation.

Our experimental design does not allow us to rule out that collagen-induced intra-dimer phosphorylation occurs. In fact, one might speculate that for closely apposed DDR1 dimers in signalling clusters, there is no functional difference between kinase domains that are linked via the transmembrane domain and those that are not. Due to the unusually large and unstructured intracellular juxtamembrane region, it is conceivable that within a signalling cluster all DDR1 kinase domains are at equal distance from one another.

[Editors' note: further revisions were requested prior to acceptance, as described below.]

*The manuscript has been improved but there are some remaining issues that need to be addressed before acceptance.*

*Your figure provided to the reviewers demonstrates that DDR function prevents formation of larger aggregates. This figure should be included into the manuscript to underline the functional relevance of your findings.*

The revised manuscript now contains the requested figure as new Figure 10, as well as a text paragraph describing the results. David Corcoran performed the experiments for this figure and is now listed as an additional author.

*We also believe that activation loop phosphorylation across dimers occurs. However, your data suggest that the more efficient mechanism occurs within dimers. The weak dimer phosphorylation exclusively occurs across dimers (inter-dimer), whereas the strong monomer phosphorylation is the sum of intra and inter dimer phosphorylation, suggesting that this sum is dominated by intra-dimer phosphorylation. There is no evidence that inter-dimer phosphorylation of the activation loop is the major event or relevant. As long as phosphorylation within dimers occurs at least as efficient as across dimers, lateral association and activation loop phosphorylation is not important.*

*Since the paper focuses on Y513 phosphorylation, it is not necessary to show that lateral association is responsible for activation loop phosphorylation. However, due to the two points above (lateral association is neither shown (1) to be activating nor (2) relevant for activation loop phosphorylation.) the statement 'Our data suggest that DDRs are activated by a mechanism in which collagen binding promotes lateral association of DDR1 dimers, which induces activation loop phosphorylation and thereby releases cis-autoinhibition of the kinase domain.' should be toned down and re-phrased appropriately.*

As requested, the sentence has been re-phrased in the amended manuscript.